# Multi-Resolution Diffusion Models for Time Series Forecasting

**Lifeng Shen**[1] , **Weiyu Chen**[2] , **James T. Kwok**[2]

[1] Division of Emerging Interdisciplinary Areas, Hong Kong University of Science and Technology

[2] Department of Computer Science and Engineering, Hong Kong University of Science and Technology

`lshenae@connect.ust.hk, wchenbx@cse.ust.hk, jamesk@cse.ust.hk`

## Abstract

The diffusion model has been successfully used in many computer vision applications, such as text-guided image generation and image-to-image translation. Recently, there have been attempts on extending the diffusion model for time series data. However, these extensions are fairly straightforward and do not utilize the unique properties of time series data. As different patterns are usually exhibited at multiple scales of a time series, we in this paper leverage this multi-resolution temporal structure and propose the multi-resolution diffusion model (`mr-Diff`). By using the seasonal-trend decomposition, we sequentially extract fine-to-coarse trends from the time series for forward diffusion. The denoising process then proceeds in an easy-to-hard non-autoregressive manner. The coarsest trend is generated first. Finer details are progressively added, using the predicted coarser trends as condition variables. Experimental results on nine real-world time series datasets demonstrate that `mr-Diff` outperforms state-of-the-art time series diffusion models. It is also better than or comparable across a wide variety of advanced time series prediction models.

## 1 Introduction

Time series data are prevalent in many real-world applications. In particular, time series forecasting facilitates users to identify patterns and make predictions based on historical data. Examples include stock price prediction in finance, patient health monitoring in healthcare, machine monitoring in manufacturing, and traffic flow optimization in transportation. Over the years, significant advancements in time series analysis have been made through the development of various deep neural networks, including recurrent neural networks (Hewamalage et al., 2021), convolutional neural networks (Yue et al., 2022), and transformers (Vaswani et al., 2017).

Besides these prominent deep neural networks, the diffusion model has recently emerged as a strong generative modeling tool. It has outperformed many other generative models in areas such as image synthesis (Ho et al., 2020; Dhariwal & Nichol, 2021), video generation (Harvey et al., 2022; Blattmann et al., 2023), and multi-modal applications (Rombach et al., 2022; Saharia et al., 2022). Very recently, researchers have sought to leverage its strong generative capacity in the time-series domain. A number of time-series diffusion models have been developed (Rasul et al., 2021; Tashiro et al., 2021; Alcaraz & Strodthoff, 2022; Shen & Kwok, 2023). For example, TimeGrad (Rasul et al., 2021) integrates the standard diffusion model with recurrent neural network's hidden states. CSDI (Tashiro et al., 2021) uses self-supervised masking to guide an non-autoregressive denoising process. While these time series diffusion models have demonstrated their efficacy, they do not fully utilize the unique structural properties in time series data and are still constrained to generate the time series directly from random vectors (Figure 1(a)). This can pose a significant challenge when working with real-world time series that are non-stationary and noisy.

As time series usually exhibit complex patterns over multiple scales, using the underlying multi-resolution temporal structure has been a cornerstone in traditional time series analysis. In particular, the seasonal-trend decomposition (Robert et al., 1990) can extract the seasonal and trend components, and the coarser temporal patterns can be used to help modeling the finer patterns. Recently, some deep time series prediction models (Oreshkin et al., 2019; Wu et al., 2021; Zeng et al., 2023) have also

incorporated multi-resolution analysis techniques. For example, NBeats (Oreshkin et al., 2019) uses the Fourier and polynomial basis to approximate the seasonal and trend components in multiple layers, respectively. Autoformer (Wu et al., 2021) uses average pooling to extract seasonal components in various transformer layers. DLinear (Zeng et al., 2023) introduces an MLP with seasonal-trend branches. N-Hits (Challu et al., 2023) uses hierarchical interpolation to better leverage the multiscale temporal patterns. Fedformer (Zhou et al., 2022b) improves Autoformer by using mixture-of-experts decomposition in the frequency domain. While these recent transformer and MLP models demonstrate the effectiveness of multi-resolution analysis in deep time series modeling, the use of multi-resolution analysis in time series diffusion models has yet to be explored.

In this paper, we bridge this gap by proposing the multi-resolution diffusion (mr-Diff) model for time series forecasting. Unlike existing time series diffusion models that directly denoises from random vectors (Figure 1(a)), mr-Diff decomposes the denoising objective into several sub-objectives (Figure 1(b)), each of which corresponds to a trend extracted from a sequence of fine-to-coarse seasonal-trend decompositions. This encourages the denoising process to proceed in an easy-to-hard manner. The coarser trends are generated first and the finer details are then progressively added. By better exploiting the seasonal-trend structure and different temporal resolutions, this leads to a more accurate generation of the time series.

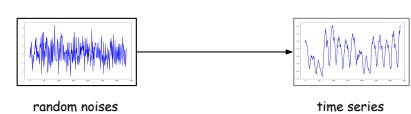

(a) *Standard diffusion with direct denoising.*

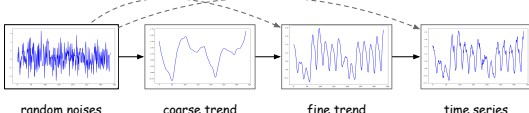

(b) *Proposed diffusion model with multi-resolution.*

Figure 1: Direct denoising versus proposed multi-resolution denoising.

The contributions of this paper can be summarized as follows: (i) We propose the multi-resolution diffusion (mr-Diff) model, which is the first to integrate the seasonal-trend decomposition-based multi-resolution analysis into time series diffusion models. (ii) We perform progressive denoising in an easy-to-hard manner, generating coarser signals first and then finer details. This allows a more accurate prediction of time series. (iii) Extensive experiments show that mr-Diff outperforms state-of-the-art time series diffusion models. It is also better than or comparable across a wide variety of advanced time series prediction models.

## 2 RELATED WORKS: DEEP TIME SERIES MODELS

Recently, a number of deep time series models have been proposed that use the transformer (Vaswani et al., 2017) to capture temporal dependencies. Informer (Zhou et al., 2021) improves the vanilla transformer by avoiding its quadratic time complexity with sparse attention, and improves the inference speed by decoding in a non-autoregressive manner. Autoformer (Wu et al., 2021) replaces the transformer's self-attention block with an auto-correlation layer. Fedformer (Zhou et al., 2022b) uses a frequency-enhanced module to capture important temporal structures by frequency domain mapping. Pyraformer (Liu et al., 2021) uses a pyramidal attention module for multi-resolution representation of the time series. Scaleformer (Shabani et al., 2023) generates the forecast progressively, starting from the coarser level and then progressing to the finer levels. PatchTST (Nie et al., 2022) is similar to the vision transformer (ViT) (Dosovitskiy et al., 2020), and performs time series prediction by patching the time series and self-supervised pre-training to extract local semantic information. It also replaces the transformer's decoder with a linear mapping and uses a channel-independence strategy to achieve good performance in multivariate time series prediction.

Besides the transformer-based models, some recent models leverage basis expansion to decompose the time series. FiLM (Zhou et al., 2022a) uses Legendre Polynomials projections to approximate historical information and Fourier projections to remove noise. NBeats (Oreshkin et al., 2019) represents the trends in the time series by polynomial coefficients and the seasonal patterns by Fourier coefficients. Depts (Fan et al., 2022) improves NBeats by using a periodicity module to model periodic time series. N-Hits (Challu et al., 2023) uses multi-scale hierarchical interpolations to further

improve NBeats. In general, these models are easier to train than transformer-based models, though their performance can vary depending on the choice of the basis.

Besides these two types of deep learning models, other recent models are also very competitive. SCINet (Liu et al., 2022) uses a recursive downsample-convolve-interact architecture to extract temporal features from the downsampled sub-sequences or features. NLinear (Zeng et al., 2023) normalizes the time series and uses a linear layer for prediction. DLinear (Zeng et al., 2023) follows the Autoformer and uses seasonal-trend decomposition.

Very recently, diffusion models have also been developed for time series data. TimeGrad (Rasul et al., 2021) is a conditional diffusion model which predicts in an autoregressive manner, with the denoising process guided by the hidden state of a recurrent neural network. However, it suffers from slow inference on long time series because of the use of autoregressive decoding. To alleviate this problem, CSDI (Tashiro et al., 2021) uses non-autoregressive generation, and uses self-supervised masking to guide the denoising process. However, it needs two transformers to capture dependencies in the channel and time dimensions. Moreover, its complexity is quadratic in the number of variables and length of time series as in other transformer models. Besides, masking-based conditioning is similar to the task of image inpainting, and can cause disharmony at the boundaries between the masked and observed regions (Lugmayr et al., 2022; Shen & Kwok, 2023). SSSD (Alcaraz & Strodthoff, 2022) reduces the computational complexity of CSDI by replacing the transformers with a structured state space model. However, it uses the same masking-based conditioning as in CSDI, and thus still suffers from the problem of boundary disharmony. To alleviate this problem, the non-autoregressive diffusion model TimeDiff (Shen & Kwok, 2023) uses future mixup and autoregressive initialization for conditioning. However, all these time series diffusion models do not leverage the multi-resolution temporal structures and denoise directly from random vectors as in standard diffusion models.

In this paper, we propose to decompose the time series into multiple resolutions using seasonal-trend decomposition, and use the fine-to-coarse trends as intermediate latent variables to guide the denoising process. Recently, other multiresolution analysis techniques besides using seasonal-trend decomposition have also been used for time series modeling. For example, Yu et al. (2021) propose a U-Net (Ronneberger et al., 2015) for graph-structured time series, and leverage temporal information from different resolutions by pooling and unpooling. Mu2ReST (Niu et al., 2022) works on spatio-temporal data and recursively outputs predictions from coarser to finer resolutions. Yformer (Madhusudhanan et al., 2021) captures temporal dependencies by combining downscaling/upsampling with sparse attention. PSA-GAN (Jeha et al., 2022) trains a growing U-Net, and captures multi-resolution patterns by progressively adding trainable modules at different levels. However, all these methods need to design very specific U-Net structures.

## 3 BACKGROUND

### 3.1 DENOISING DIFFUSION PROBABILISTIC MODELS

A well-known diffusion model is the denoising diffusion probabilistic model (DDPM) (Ho et al., 2020). It is a latent variable model with forward diffusion and backward denoising processes. During forward diffusion, an input $\mathbf{x}^0$ is gradually corrupted to a Gaussian noise vector. Specifically, at the $k$th step, $\mathbf{x}^k$ is generated by corrupting the previous iterate $\mathbf{x}^{k-1}$ (scaled by $\sqrt{1-\beta_k}$) with zero-mean Gaussian noise (with variance $\beta_k \in [0, 1]$):

$$q(\mathbf{x}^k|\mathbf{x}^{k-1}) = \mathcal{N}(\mathbf{x}^k; \sqrt{1-\beta_k}\mathbf{x}^{k-1}, \beta_k\mathbf{I}), \quad k = 1, \ldots, K.$$

It can be shown that this can also be rewritten as $q(\mathbf{x}^k|\mathbf{x}^0) = \mathcal{N}(\mathbf{x}^k; \sqrt{\bar{\alpha}_k}\mathbf{x}^0, (1 - \bar{\alpha}_k)\mathbf{I})$, where $\bar{\alpha}_k = \Pi_{s=1}^k \alpha_s$, and $\alpha_k = 1 - \beta_k$. Thus, $\mathbf{x}^k$ can be simply obtained as

$$\mathbf{x}^k = \sqrt{\bar{\alpha}_k}\mathbf{x}^0 + \sqrt{1 - \bar{\alpha}_k}\epsilon, \tag{1}$$

where $\epsilon$ is a noise from $\mathcal{N}(\mathbf{0}, \mathbf{I})$. This equation also allows $\mathbf{x}^0$ to be easily recovered from $\mathbf{x}^k$.

In DDPM, backward denoising is defined as a Markovian process. Specifically, at the $k$th denoising step, $\mathbf{x}^{k-1}$ is generated from $\mathbf{x}^k$ by sampling from the following normal distribution:

$$p_\theta(\mathbf{x}^{k-1}|\mathbf{x}^k) = \mathcal{N}(\mathbf{x}^{k-1}; \mu_\theta(\mathbf{x}^k, k), \Sigma_\theta(\mathbf{x}^k, k)). \tag{2}$$

Here, the variance $\Sigma_\theta(\mathbf{x}^k, k)$ is usually fixed as $\sigma_k^2\mathbf{I}$, while the mean $\mu_\theta(\mathbf{x}^k, k)$ is defined by a neural network (parameterized by $\theta$). This is usually formulated as a noise estimation or data prediction problem (Benny & Wolf, 2022). For noise estimation, a network $\epsilon_\theta$ predicts the noise of the diffused input $\mathbf{x}^k$, and then obtains $\mu_\theta(\mathbf{x}^k, k)$ as $\frac{1}{\sqrt{\alpha_k}}\mathbf{x}^k - \frac{1-\alpha_k}{\sqrt{1-\bar\alpha_k}\sqrt{\alpha_k}}\epsilon_\theta(\mathbf{x}^k, k)$. Parameter $\theta$ is learned by minimizing the loss $\mathcal{L}_\epsilon = \mathbb{E}_{k,\mathbf{x}^0,\epsilon}\left[\|\epsilon - \epsilon_\theta(\mathbf{x}^k, k)\|^2\right]$. Alternatively, the data prediction strategy uses a denoising network $\mathbf{x}_\theta$ to obtain an estimate $\mathbf{x}_\theta(\mathbf{x}^k, k)$ of the clean data $\mathbf{x}^0$ given $\mathbf{x}^k$, and then set

$$\mu_\theta(\mathbf{x}^k, k) = \frac{\sqrt{\alpha_k}(1-\bar\alpha_{k-1})}{1-\bar\alpha_k}\mathbf{x}^k + \frac{\sqrt{\bar\alpha_{k-1}}\beta_k}{1-\bar\alpha_k}\mathbf{x}_\theta(\mathbf{x}^k, k). \tag{3}$$

Parameter $\theta$ is learned by minimizing the loss

$$\mathcal{L}_\mathbf{x} = \mathbb{E}_{\mathbf{x}^0,\epsilon,k}\|\mathbf{x}^0 - \mathbf{x}_\theta(\mathbf{x}^k, k)\|^2. \tag{4}$$

## 3.2 Conditional Diffusion Models for Time Series Prediction

In time series forecasting, one aims to predict the future values $\mathbf{x}_{1:H}^0 \in \mathbb{R}^{d \times H}$ given the past observations $\mathbf{x}_{-L+1:0}^0 \in \mathbb{R}^{d \times L}$ of the time series. Here, $d$ is the number of variables, $H$ is the length of the forecast window, and $L$ is the length of the lookback window. When using conditional diffusion models for time series prediction, the following distribution is considered (Rasul et al., 2021; Tashiro et al., 2021; Shen & Kwok, 2023)

$$p_\theta(\mathbf{x}_{1:H}^{0:K}|\mathbf{c}) = p_\theta(\mathbf{x}_{1:H}^K)\prod_{k=1}^K p_\theta(\mathbf{x}_{1:H}^{k-1}|\mathbf{x}_{1:H}^k, \mathbf{c}), \quad \mathbf{c} = \mathcal{F}(\mathbf{x}_{-L+1:0}^0), \tag{5}$$

where $\mathbf{x}_{1:H}^K \sim \mathcal{N}(\mathbf{0}, \mathbf{I})$, $\mathbf{c}$ is the condition, and $\mathcal{F}$ is a conditioning network that takes the past observations $\mathbf{x}_{-L+1:0}^0$ as input. Correspondingly, the denoising process at step $k$ is given by

$$p_\theta(\mathbf{x}_{1:H}^{k-1}|\mathbf{x}_{1:H}^k, \mathbf{c}) = \mathcal{N}(\mathbf{x}_{1:H}^{k-1}; \mu_\theta(\mathbf{x}_{1:H}^k, k|\mathbf{c}), \sigma_k^2\mathbf{I}), \quad k = K, K-1, \ldots, 1. \tag{6}$$

During inference, we denote the generated sample corresponding to $\mathbf{x}_{1:H}^k$ by $\hat{\mathbf{x}}_{1:H}^k$. We first initialize $\hat{\mathbf{x}}_{1:H}^K$ as a noise vector from $\mathcal{N}(\mathbf{0}, \mathbf{I})$. By repeatedly running the denoising step in (6) till $k = 1$, the final generated sample is $\hat{\mathbf{x}}_{1:H}^0$.

## 4 mr-Diff: Multi-Resolution Diffusion Model

As discussed in Section 1, recent transformer and MLP models demonstrate the effectiveness of seasonal-trend decomposition-based multi-resolution analysis in deep time series modeling. However, the use of multi-resolution temporal patterns in the diffusion model has yet to be explored. In this paper, we address this gap by proposing the multi-resolution diffusion (mr-Diff) model. An overview of the proposed model is shown in Figure 2.

The proposed mr-Diff can be viewed as a cascaded diffusion model (Ho et al., 2022), and proceeds in $S$ stages, with the resolution getting coarser as the stage proceeds (Section 4.1). This allows capturing the temporal dynamics at multiple temporal resolutions. In each stage, the diffusion process is interleaved with seasonal-trend decomposition. For simplicity of notations, we use $\mathbf{X} = \mathbf{x}_{-L+1:0}$ and $\mathbf{Y} = \mathbf{x}_{1:H}$ for the time series segments in the lookback and forecast windows, respectively. Let the trend component of the lookback (resp. forecast) segment at stage $s+1$ be $\mathbf{X}_s$ (resp. $\mathbf{Y}_s$). The trend gets coarser as $s$ increases, and with $\mathbf{X}_0 = \mathbf{X}$ and $\mathbf{Y}_0 = \mathbf{Y}$. In each stage $s+1$, a conditional diffusion model is learned to reconstruct the trend component $\mathbf{Y}_s$ extracted from the forecast window (Section 4.2). The reconstruction at stage 1 then corresponds to the target time series forecast.

While the forward diffusion process in this diffusion model is straightforward and similar to existing diffusion models, design of the denoising process, particularly on the denoising conditions and denoising network, are less trivial. During training, to guide the reconstruction of $\mathbf{Y}_s$, the proposed model takes the lookback segment $\mathbf{X}_s$ (which has the same resolution as $\mathbf{Y}_s$) and the coarser trend $\mathbf{Y}_{s+1}$ (which provides an overall picture of the finer $\mathbf{Y}_s$) as denoising condition. On inference, the ground-truth $\mathbf{Y}_{s+1}$ is not available, and is replaced by its estimate $\hat{\mathbf{Y}}_{s+1}^0$ produced by the denoising process at stage $s+1$. By combining the diffusion model and seasonal-trend decomposition in a multi-resolution manner, the proposed model encourages a better modeling of real-world time series.

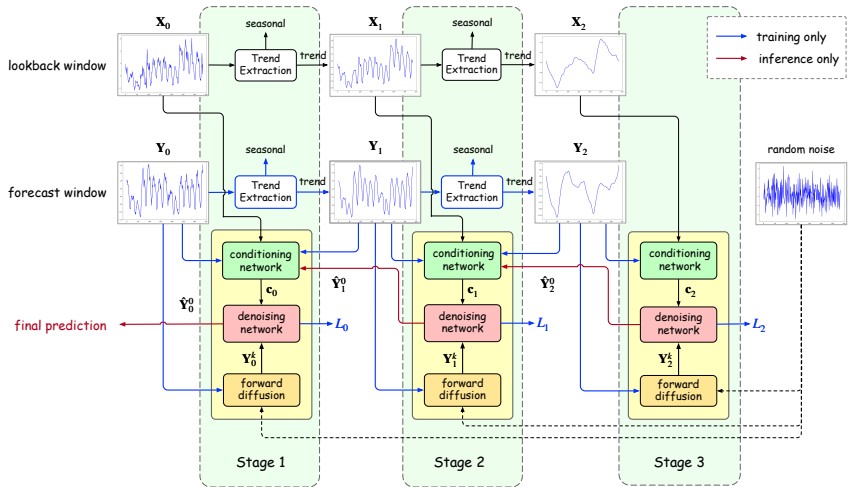

Figure 2: The proposed multi-resolution diffusion model `mr-Diff`. For simplicity of illustration, we use $S = 3$ stages. At stage $s + 1$, $\mathbf{Y}_s$ denotes the corresponding trend component extracted from the segment in the forecast window, $\mathbf{Y}_s^k$ is the diffusion sample at diffusion step $k$, and $\hat{\mathbf{Y}}_s^0$ is the denoised output.

## 4.1 EXTRACTING FINE-TO-COARSE TRENDS

For the given time series segment $\mathbf{X}_0$ in the lookback window, its trend components are successively extracted by the `TrendExtraction` module as:

$$\mathbf{X}_s = \mathtt{AvgPool}(\mathtt{Padding}(\mathbf{X}_{s-1}), \tau_s), \quad s = 1, \ldots, S - 1,$$

where `AvgPool` is the average pooling operation (Wu et al., 2021), `Padding` keeps the lengths of $\mathbf{X}_{s-1}$ and $\mathbf{X}_s$ the same, and $\tau_s$ is the smoothing kernel size which increases with $s$ so as to generate fine-to-coarse trends. Processing for the segment $\mathbf{Y}_0$ in the forecast window is analogous, and its trend components $\{\mathbf{Y}_s\}_{s=1,\ldots,S-1}$ are extracted.

Note that while the seasonal-trend decomposition obtains both the seasonal and trend components, the focus here is on the trend. On the other hand, models such as the Autoformer (Wu et al., 2021) and Fedformer (Zhou et al., 2022b) focus on progressively decomposing the seasonal component. As we use the diffusion model for time series reconstruction at various stages/resolutions (Section 4.2), intuitively, it is easier to predict a finer trend from a coarser trend. On the other hand, reconstruction of a finer seasonal component from a coarser seasonal component may be difficult, especially as the seasonal component may not present clear patterns.

## 4.2 TEMPORAL MULTI-RESOLUTION RECONSTRUCTION

In each stage $s + 1$, we use a conditional diffusion model to reconstruct the future trend $\mathbf{Y}_s$ extracted in Section 4.1. As in the standard diffusion model, it consists of a forward diffusion process and a backward denoising process. Forward diffusion does not involve learnable parameters, while the training procedure of the backward denoising process needs optimization as shown in Algorithm 1.

An $d'$-dimensional embedding $\mathbf{p}^k$ of the diffusion step $k$ is used in both the forward and backward denoising processes. As in (Rasul et al., 2021; Tashiro et al., 2021; Kong et al., 2020), this is obtained by first taking the sinusoidal position embedding (Vaswani et al., 2017): $k_{\mathtt{embedding}} = \left[\sin(10^{\frac{0 \times 4}{w-1}} t), \ldots, \sin(10^{\frac{w \times 4}{w-1}} t), \cos(10^{\frac{0 \times 4}{w-1}} t), \ldots, \cos(10^{\frac{w \times 4}{w-1}} t)\right]$ where $w = \frac{d'}{2}$, and then passing it through two fully-connected (FC) layers to obtain

$$\mathbf{p}^k = \mathrm{SiLU}(\mathrm{FC}(\mathrm{SiLU}(\mathrm{FC}(k_{\mathtt{embedding}})))), \tag{7}$$

where SiLU is the sigmoid-weighted linear unit (Elfwing et al., 2017). By default, $d'$ is set to 128.

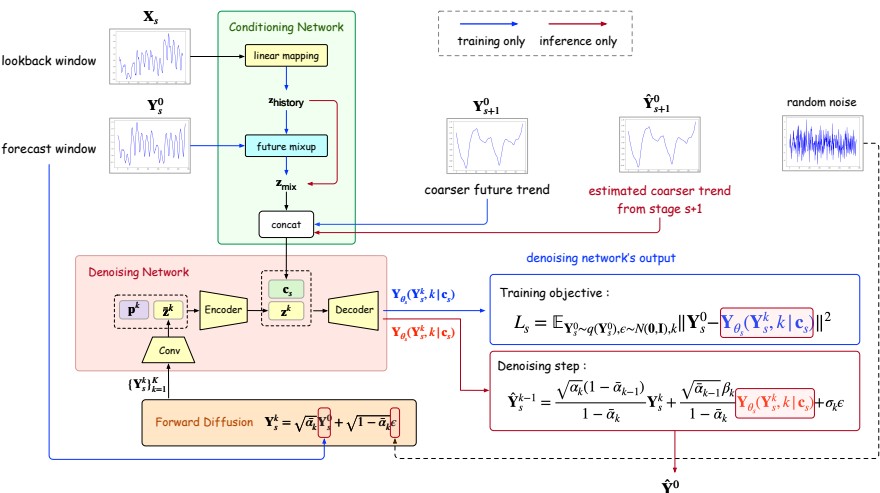

Figure 3: The conditioning network and denoising network.

### 4.2.1 FORWARD DIFFUSION

Forward diffusion is straightforward. Analogous to (1), with $\mathbf{Y}_s^0 = \mathbf{Y}_s$, we obtain at step $k$,

$$\mathbf{Y}_s^k = \sqrt{\bar{\alpha}_k}\mathbf{Y}_s^0 + \sqrt{1-\bar{\alpha}_k}\epsilon, \quad k = 1, \ldots, K, \tag{8}$$

where the noise matrix $\epsilon$ is sampled from $\mathcal{N}(\mathbf{0}, \mathbf{I})$ with the same size as $\mathbf{Y}_s$.

### 4.2.2 BACKWARD DENOISING

Standard diffusion models perform one-stage denoising directly from random vectors. In this section, we decompose the denoising objective of mr-Diff, into $S$ sub-objectives, one for each stage. As will be seen, this encourages the denoising process to proceed in an easy-to-hard manner, such that the coarser trends are generated first and the finer details are then progressively added.

**Conditioning network.** The conditioning network (Figure 3, top) constructs a condition to guide the denoising network (to be discussed). Existing time series diffusion models (Rasul et al., 2021; Tashiro et al., 2021) simply use the original time series' lookback segment $\mathbf{X}_0$ as condition $\mathbf{c}$ in (5). With the proposed multi-resolution seasonal-trend decomposition, we instead use the lookback segment $\mathbf{X}_s$ at the same decomposition stage $s$. This allows better and easier reconstruction, as $\mathbf{X}_s$ has the same resolution as $\mathbf{Y}_s$ to be reconstructed. On the other hand, when $\mathbf{X}_0$ is used as in existing time series diffusion models, the denoising network may overfit temporal details at the finer level.

A linear mapping is applied on $\mathbf{X}_s$ to produce a tensor $\mathbf{z}_{\texttt{history}} \in \mathbb{R}^{d \times H}$. For better denoising performance, during training we use future-mixup (Shen & Kwok, 2023) to enhance $\mathbf{z}_{\texttt{history}}$. It combines $\mathbf{z}_{\texttt{history}}$ and the (ground-truth) future observation $\mathbf{Y}_s^0$ with mixup (Zhang et al., 2018):

$$\mathbf{z}_{\texttt{mix}} = \mathbf{m} \odot \mathbf{z}_{\texttt{history}} + (1-\mathbf{m}) \odot \mathbf{Y}_s^0, \tag{9}$$

where $\odot$ denotes the Hadamard product, and $\mathbf{m} \in [0,1)^{d \times H}$ is a mixing matrix in which each element is randomly sampled from the uniform distribution on $[0,1)$. Future-mixup is similar to teacher forcing (Williams & Zipser, 1989), which mixes the ground truth with previous prediction output at each decoding step of autoregressive decoding.

Besides using $\mathbf{X}_s$ on training, the coarser trend $\mathbf{Y}_{s+1}$ ($= \mathbf{Y}_{s+1}^0$) can provide an overall picture of the finer trend $\mathbf{Y}_s$ and is thus also useful for conditioning. Hence, $\mathbf{z}_{\texttt{mix}}$ in (9) is concatenated with $\mathbf{Y}_{s+1}^0$ along the channel dimension to produce the condition $\mathbf{c}_s$ (a $2d \times H$ tensor). For $s = S$ (the last stage), there is no coarser trend and $\mathbf{c}_s$ is simply equal to $\mathbf{z}_{\texttt{mix}}$.

On inference, the ground-truth $\mathbf{Y}_s^0$ is no longer available. Hence, future-mixup in (9) is not used, and we simply set $\mathbf{z}_{\texttt{mix}} = \mathbf{z}_{\texttt{history}}$. Moreover, the coarser trend $\mathbf{Y}_{s+1}$ is also not available, and we concatenate $\mathbf{z}_{\texttt{mix}}$ with the estimate $\hat{\mathbf{Y}}_{s+1}^0$ generated from stage $s + 2$ instead.

**Denoising network.** Analogous to (6), the denoising process at step $k$ of stage $s + 1$ is given by

$$p_{\theta_s}(\mathbf{Y}_s^{k-1}|\mathbf{Y}_s^k, \mathbf{c}_s) = \mathcal{N}(\mathbf{Y}_s^{k-1}; \mu_{\theta_s}(\mathbf{Y}_s^k, k|\mathbf{c}_s, \sigma_k^2\mathbf{I})), \quad k = K, \ldots, 1, \tag{10}$$

where $\theta_s$ includes all parameters in the conditional network and denoising network at stage $s + 1$, and

$$\mu_{\theta_s}(\mathbf{Y}_s^k, k|\mathbf{c}_s, \sigma_k^2\mathbf{I}) = \frac{\sqrt{\alpha_k}(1-\bar{\alpha}_{k-1})}{1-\bar{\alpha}_k}\mathbf{Y}_s^k + \frac{\sqrt{\bar{\alpha}_{k-1}}\beta_k}{1-\bar{\alpha}_k}\mathbf{Y}_{\theta_s}(\mathbf{Y}_s^k, k|\mathbf{c}_s). \tag{11}$$

As in (3), $\mathbf{Y}_{\theta_s}(\mathbf{Y}_s^k, k|\mathbf{c}_s)$ is an estimate of $\mathbf{Y}_s^0$.

The denoising network (also shown in Figure 3) outputs $\mathbf{Y}_{\theta_s}(\mathbf{Y}_s^k, k|\mathbf{c}_s)$ with guidance from the condition $\mathbf{c}_s$ (output by the conditioning network). Specifically, it first maps $\mathbf{Y}_s^k$ to the embedding $\bar{\mathbf{z}}^k \in \mathbb{R}^{d' \times H}$ by an input projection block consisting of several convolutional layers. This $\bar{\mathbf{z}}^k$, together with the diffusion-step $k$'s embedding $\mathbf{p}^k \in \mathbb{R}^{d'}$ in (7), is fed to an encoder (which is a convolution network) to obtain the representation $\mathbf{z}^k \in \mathbb{R}^{d'' \times H}$. Next, we concatenate $\mathbf{z}^k$ and $\mathbf{c}_s$ along the variable dimension to form a tensor of size $(2d + d'') \times H$. This is then fed to a decoder, which is also a convolution network, and outputs $\mathbf{Y}_{\theta_s}(\mathbf{Y}_s^k, k|\mathbf{c}_s)$. Finally, analogous to (4), $\theta_s$ is obtained by minimizing the following denoising objective

$$\min_{\theta_s} \mathcal{L}_s(\theta_s) = \min_{\theta_s} \mathbb{E}_{\mathbf{Y}_s^0 \sim q(\mathbf{Y}_s), \epsilon \sim \mathcal{N}(\mathbf{0}, \mathbf{I}), k} \left\| \mathbf{Y}_s^0 - \mathbf{Y}_{\theta_s}(\mathbf{Y}_s^k, k|\mathbf{c}_s) \right\|^2. \tag{12}$$

On inference, for each $s = S, \ldots, 1$, we start from $\hat{\mathbf{Y}}_s^K \sim \mathcal{N}(\mathbf{0}, \mathbf{I})$ (step 9 in Algorithm 2). Based on the data prediction strategy in (11), each denoising step from $\hat{\mathbf{Y}}_s^k$ (an estimate of $\mathbf{Y}_s^k$) to $\hat{\mathbf{Y}}_s^{k-1}$ is:

$$\hat{\mathbf{Y}}_s^{k-1} = \frac{\sqrt{\alpha_k}(1-\bar{\alpha}_{k-1})}{1-\bar{\alpha}_k}\hat{\mathbf{Y}}_s^k + \frac{\sqrt{\bar{\alpha}_{k-1}}\beta_k}{1-\bar{\alpha}_k}\mathbf{Y}_{\theta_s}(\hat{\mathbf{Y}}_s^k, k|\mathbf{c}_s) + \sigma_k\epsilon, \tag{13}$$

where $\epsilon \sim \mathcal{N}(\mathbf{0}, \mathbf{I})$ when $k > 1$, and $\epsilon = 0$ otherwise.

The pseudocode for the training and inference procedures of the backward denoising process can be found in Appendix A.

**Discussion.** The proposed `mr-Diff` is related to the Scaleformer (Shabani et al., 2023), which also generates the forecast progressively from the coarser level to the finer ones. However, besides the obvious difference that Scaleformer is based on the transformer while `mr-Diff` is based on the diffusion model, Scaleformer uses one single forecasting module (a transformer) at all resolutions. However, as the time series patterns at different resolutions may be different, in `mr-Diff`, each resolution has its own denoising network and is thus more flexible. Moreover, in moving from a lower resolution to a higher resolution during both training and inference, Scaleformer needs linear interpolation and an additional cross-scale normalization mechanism. On the other hand, in `mr-Diff`, the lower-resolution prediction is fed into the conditioning network as a condition. The conditioning network can learn a more flexible nonlinear mapping to fuse the cross-scale trends. Coherent probabilistic forecasting (Rangapuram et al., 2023) also uses a temporal hierarchy to produce forecasts at multiple resolutions. However, it is more stringent than `mr-Diff` and requires the model to produce consistent forecasts at all resolutions. As will be shown in Section 5, this enables `mr-Diff` to have better empirical performance.

## 5 EXPERIMENTS

In this section, we conduct time series prediction experiments by comparing 22 recent strong prediction models on 9 popular real-world time series datasets. Due to space constraints, detailed introductions about the datasets and selected baselines are in Appendix B and Appendix C, respectively. For performance evaluation, we use the mean absolute error (MAE) and mean squared error (MSE) averaged over ten randomly sampled trajectories. Because of the lack of space, results on MSE are in Appendix D. Furthermore, ablation studies are provided in Appendix E.

**Implementation Details.** We train the proposed model using Adam with a learning rate of $10^{-3}$. The batch size is 64, and training with early stopping for a maximum of 100 epochs. $K = 100$ diffusion steps are used, with a linear variance schedule (Rasul et al., 2021) starting from $\beta_1 = 10^{-4}$ to $\beta_K = 10^{-1}$. $S = 5$ stages are used. The history length (in {96, 192, 336, 720, 1440}) is selected by using the validation set. All experiments are run on an Nvidia RTX A6000 GPU with 48GB memory. More details can be found in Appendix F.

Table 1: Univariate prediction MAEs on the real-world time series datasets (subscript is the rank). Results of all baselines (except NLinear, SCINet, and N-Hits) are from (Shen & Kwok, 2023).

| | NorPool | Caiso | Traffic | Electricity | Weather | Exchange | ETTh1 | ETTm1 | Wind | avg rank |
|---|---|---|---|---|---|---|---|---|---|---|
| **mr-Diff** | $\underline{0.609}_{(2)}$ | $0.212_{(4)}$ | $\underline{0.197}_{(2)}$ | $\mathbf{0.332}_{(1)}$ | $\mathbf{0.032}_{(1)}$ | $\mathbf{0.094}_{(1)}$ | $\mathbf{0.196}_{(1)}$ | $\mathbf{0.149}_{(1)}$ | $\underline{1.168}_{(2)}$ | 1.7 |
| TimeDiff | $0.613_{(3)}$ | $0.209_{(3)}$ | $0.207_{(3)}$ | $0.341_{(3)}$ | $0.035_{(4)}$ | $0.102_{(7)}$ | $\underline{0.202}_{(2)}$ | $0.154_{(6)}$ | $1.209_{(5)}$ | 4.0 |
| TimeGrad | $0.841_{(23)}$ | $0.386_{(22)}$ | $0.894_{(23)}$ | $0.898_{(23)}$ | $0.036_{(6)}$ | $0.155_{(21)}$ | $0.212_{(8)}$ | $0.167_{(12)}$ | $1.239_{(11)}$ | 16.6 |
| CSDI | $0.763_{(20)}$ | $0.282_{(14)}$ | $0.468_{(20)}$ | $0.540_{(19)}$ | $0.037_{(7)}$ | $0.200_{(23)}$ | $0.221_{(12)}$ | $0.170_{(14)}$ | $1.218_{(7)}$ | 15.1 |
| SSSD | $0.770_{(21)}$ | $0.263_{(12)}$ | $0.226_{(6)}$ | $0.403_{(8)}$ | $0.041_{(11)}$ | $0.118_{(16)}$ | $0.250_{(20)}$ | $0.169_{(13)}$ | $1.356_{(22)}$ | 14.3 |
| D³VAE | $0.774_{(22)}$ | $0.613_{(23)}$ | $0.237_{(9)}$ | $0.539_{(18)}$ | $0.039_{(9)}$ | $0.107_{(13)}$ | $0.221_{(12)}$ | $0.160_{(9)}$ | $1.321_{(19)}$ | 14.9 |
| CPF | $0.710_{(15)}$ | $0.338_{(18)}$ | $0.385_{(19)}$ | $0.592_{(21)}$ | $0.035_{(4)}$ | $0.094_{(1)}$ | $0.221_{(12)}$ | $0.153_{(5)}$ | $1.256_{(12)}$ | 11.9 |
| PSA-GAN | $0.623_{(5)}$ | $0.250_{(8)}$ | $0.355_{(17)}$ | $0.373_{(6)}$ | $0.139_{(22)}$ | $0.109_{(14)}$ | $0.225_{(16)}$ | $0.174_{(16)}$ | $1.287_{(16)}$ | 13.3 |
| N-Hits | $0.646_{(7)}$ | $0.276_{(13)}$ | $0.232_{(7)}$ | $0.419_{(9)}$ | $\underline{0.033}_{(2)}$ | $0.100_{(5)}$ | $0.228_{(17)}$ | $0.157_{(8)}$ | $1.256_{(12)}$ | 8.9 |
| FiLM | $0.654_{(9)}$ | $0.290_{(14)}$ | $0.315_{(15)}$ | $0.362_{(5)}$ | $0.069_{(15)}$ | $0.104_{(10)}$ | $0.210_{(6)}$ | $\mathbf{0.149}_{(1)}$ | $1.189_{(3)}$ | 8.6 |
| Depts | $0.616_{(4)}$ | $\mathbf{0.205}_{(1)}$ | $0.241_{(10)}$ | $0.434_{(12)}$ | $0.102_{(19)}$ | $0.106_{(12)}$ | $\underline{0.202}_{(2)}$ | $0.165_{(10)}$ | $1.472_{(23)}$ | 10.3 |
| NBeats | $0.671_{(10)}$ | $0.228_{(5)}$ | $0.225_{(5)}$ | $0.439_{(13)}$ | $0.130_{(21)}$ | $\underline{0.096}_{(3)}$ | $0.242_{(18)}$ | $0.165_{(10)}$ | $1.236_{(9)}$ | 10.4 |
| Scaleformer | $0.687_{(12)}$ | $0.320_{(16)}$ | $0.375_{(18)}$ | $0.430_{(10)}$ | $0.083_{(17)}$ | $0.148_{(19)}$ | $0.302_{(22)}$ | $0.210_{(22)}$ | $1.348_{(21)}$ | 17.4 |
| PatchTST | $\mathbf{0.590}_{(1)}$ | $0.260_{(11)}$ | $0.269_{(11)}$ | $0.478_{(17)}$ | $0.098_{(18)}$ | $0.111_{(15)}$ | $0.260_{(21)}$ | $0.174_{(16)}$ | $1.338_{(20)}$ | 14.4 |
| FedFormer | $0.725_{(17)}$ | $0.254_{(9)}$ | $0.278_{(12)}$ | $0.453_{(14)}$ | $0.057_{(13)}$ | $0.168_{(22)}$ | $0.212_{(8)}$ | $0.195_{(20)}$ | $1.271_{(14)}$ | 14.3 |
| Autoformer | $0.755_{(19)}$ | $0.339_{(19)}$ | $0.495_{(21)}$ | $0.623_{(22)}$ | $0.040_{(10)}$ | $0.152_{(20)}$ | $0.220_{(11)}$ | $0.174_{(16)}$ | $1.319_{(18)}$ | 17.3 |
| Pyraformer | $0.747_{(18)}$ | $0.257_{(10)}$ | $0.215_{(4)}$ | $0.455_{(15)}$ | $0.107_{(20)}$ | $0.104_{(10)}$ | $0.211_{(7)}$ | $0.179_{(19)}$ | $1.284_{(15)}$ | 13.1 |
| Informer | $0.698_{(13)}$ | $0.345_{(20)}$ | $0.308_{(13)}$ | $0.433_{(11)}$ | $0.069_{(14)}$ | $0.118_{(16)}$ | $0.212_{(8)}$ | $0.172_{(15)}$ | $1.236_{(9)}$ | 13.2 |
| Transformer | $0.723_{(16)}$ | $0.345_{(20)}$ | $0.336_{(16)}$ | $0.469_{(16)}$ | $0.071_{(16)}$ | $0.103_{(9)}$ | $0.247_{(19)}$ | $0.196_{(21)}$ | $1.212_{(6)}$ | 15.4 |
| SCINet | $0.653_{(8)}$ | $0.244_{(7)}$ | $0.322_{(15)}$ | $0.377_{(7)}$ | $0.037_{(7)}$ | $0.101_{(6)}$ | $0.205_{(5)}$ | $\underline{0.150}_{(4)}$ | $\mathbf{1.167}_{(1)}$ | 6.7 |
| NLinear | $0.637_{(6)}$ | $0.238_{(6)}$ | $\mathbf{0.192}_{(1)}$ | $\underline{0.334}_{(2)}$ | $\underline{0.033}_{(2)}$ | $0.097_{(4)}$ | $0.203_{(4)}$ | $\mathbf{0.149}_{(1)}$ | $1.197_{(4)}$ | 3.3 |
| DLinear | $0.671_{(10)}$ | $\underline{0.206}_{(2)}$ | $0.236_{(8)}$ | $0.348_{(4)}$ | $0.310_{(23)}$ | $0.102_{(7)}$ | $0.222_{(15)}$ | $0.155_{(7)}$ | $1.221_{(8)}$ | 9.3 |
| LSTMa | $0.707_{(14)}$ | $0.333_{(17)}$ | $0.757_{(22)}$ | $0.557_{(20)}$ | $0.053_{(12)}$ | $0.136_{(18)}$ | $0.332_{(23)}$ | $0.239_{(23)}$ | $1.298_{(17)}$ | 18.4 |

## 5.1 MAIN RESULTS

Table 1 shows the MAEs on the univariate time series. As can be seen, the proposed `mr-Diff` is the best in 5 of the 9 datasets. The improvement is particularly significant on the more complicated datasets, such as *Exchange* and *ETTh1*. On the remaining 4 datasets, `mr-Diff` ranks second in 3 of them. Overall, its average ranking is better than all other baselines (which include the most recent diffusion models). Note that there is no improvement on datasets such as *Caiso*. This is because there is no complex trend information on this dataset that can be leveraged (as can be seen in Figure 5(b) Appendix B).

Figure 4 shows example prediction results on *ETTh1* by `mr-Diff` and three competitive models: SCINet, NLinear and the recent diffusion model TimeDiff. As can be seen, `mr-Diff` produces higher-quality predictions than the others.

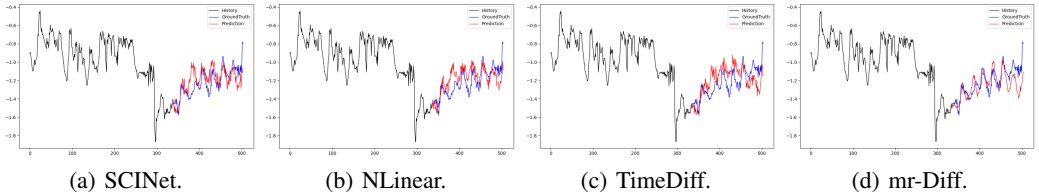

| (a) SCINet. | (b) NLinear. | (c) TimeDiff. | (d) mr-Diff. |

Figure 4: Visualizations on *ETTh1* by (a) SCINet (Liu et al., 2022), (b) NLinear (Zeng et al., 2023), (c) TimeDiff (Shen & Kwok, 2023) and (d) the proposed `mr-Diff`.

Table 2 shows the MAEs on the multivariate time series. Baselines N-Hits, FiLM and SCINet have strong performance because they also leverage multiresolution information in prediction. However, the proposed `mr-Diff` is still very competitive compared to these strong baselines. It ranks among the top-2 methods on 5 of the 9 datasets, and is the best overall. This is then followed by the recent diffusion-based model TimeDiff.

## 5.2 INFERENCE EFFICIENCY

In this experiment, we compare the inference efficiency of `mr-Diff` (with 1-4 stages) with the other time series diffusion model baselines (TimeDiff, TimeGrad, CSDI, SSSD) on the univariate *ETTh1* dataset. Due to space constraints, we compare the training efficiency in Appendix G. Besides using a

Table 2: Multivariate prediction MAEs on the real-world time series datasets (subscript is the rank). CSDI runs out of memory on *Traffic* and *Electricity*. Results of all baselines (except for NLinear, SCINet, and N-Hits) are from (Shen & Kwok, 2023).

| | NorPool | Caiso | Traffic | Electricity | Weather | Exchange | ETTh1 | ETTm1 | Wind | avg rank |
|---|---|---|---|---|---|---|---|---|---|---|
| **mr-Diff** | $0.604_{(2)}$ | $0.219_{(3)}$ | $0.320_{(5)}$ | $0.252_{(3)}$ | $0.324_{(2)}$ | $0.082_{(3)}$ | $0.422_{(2)}$ | $0.373_{(2)}$ | $0.675_{(1)}$ | 2.6 |
| TimeDiff | $0.611_{(3)}$ | $0.234_{(6)}$ | $0.384_{(8)}$ | $0.305_{(6)}$ | $0.312_{(1)}$ | $0.091_{(7)}$ | $0.430_{(3)}$ | $0.372_{(1)}$ | $0.687_{(3)}$ | 4.2 |
| TimeGrad | $0.821_{(19)}$ | $0.339_{(18)}$ | $0.849_{(22)}$ | $0.630_{(21)}$ | $0.381_{(14)}$ | $0.193_{(19)}$ | $0.719_{(22)}$ | $0.605_{(22)}$ | $0.793_{(21)}$ | 19.8 |
| CSDI | $0.777_{(17)}$ | $0.345_{(19)}$ | - | - | $0.374_{(12)}$ | $0.194_{(20)}$ | $0.438_{(5)}$ | $0.442_{(15)}$ | $0.741_{(10)}$ | 14.0 |
| SSSD | $0.753_{(13)}$ | $0.295_{(10)}$ | $0.398_{(13)}$ | $0.363_{(12)}$ | $0.350_{(8)}$ | $0.127_{(12)}$ | $0.561_{(17)}$ | $0.406_{(10)}$ | $0.778_{(18)}$ | 12.6 |
| D$^3$VAE | $0.692_{(9)}$ | $0.331_{(16)}$ | $0.483_{(17)}$ | $0.372_{(14)}$ | $0.380_{(13)}$ | $0.301_{(22)}$ | $0.502_{(14)}$ | $0.391_{(8)}$ | $0.779_{(19)}$ | 14.7 |
| CPF | $0.889_{(21)}$ | $0.424_{(21)}$ | $0.714_{(21)}$ | $0.643_{(22)}$ | $0.781_{(23)}$ | $0.082_{(3)}$ | $0.597_{(20)}$ | $0.472_{(16)}$ | $0.757_{(15)}$ | 18.0 |
| PSA-GAN | $0.890_{(22)}$ | $0.477_{(22)}$ | $0.697_{(20)}$ | $0.533_{(20)}$ | $0.578_{(22)}$ | $0.087_{(6)}$ | $0.546_{(16)}$ | $0.488_{(18)}$ | $0.756_{(13)}$ | 17.7 |
| N-Hits | $0.643_{(7)}$ | $0.221_{(4)}$ | $0.268_{(2)}$ | $0.245_{(2)}$ | $0.335_{(4)}$ | $0.085_{(5)}$ | $0.480_{(8)}$ | $0.388_{(6)}$ | $0.734_{(8)}$ | 5.1 |
| FiLM | $0.646_{(8)}$ | $0.278_{(8)}$ | $0.398_{(13)}$ | $0.320_{(8)}$ | $0.336_{(5)}$ | $0.079_{(1)}$ | $0.436_{(4)}$ | $0.374_{(3)}$ | $0.717_{(5)}$ | 6.1 |
| Depts | $0.611_{(3)}$ | $0.204_{(2)}$ | $0.568_{(19)}$ | $0.401_{(17)}$ | $0.394_{(16)}$ | $0.100_{(9)}$ | $0.491_{(12)}$ | $0.412_{(12)}$ | $0.751_{(12)}$ | 11.3 |
| NBeats | $0.832_{(20)}$ | $0.235_{(7)}$ | $0.265_{(1)}$ | $0.370_{(13)}$ | $0.420_{(17)}$ | $0.081_{(2)}$ | $0.521_{(15)}$ | $0.409_{(11)}$ | $0.741_{(10.5)}$ | 10.7 |
| Scaleformer | $0.769_{(16)}$ | $0.310_{(12)}$ | $0.379_{(7)}$ | $0.304_{(5)}$ | $0.438_{(18)}$ | $0.138_{(14)}$ | $0.579_{(19)}$ | $0.475_{(17)}$ | $0.864_{(22)}$ | 14.4 |
| PatchTST | $0.710_{(10)}$ | $0.293_{(9)}$ | $0.411_{(16)}$ | $0.348_{(11)}$ | $0.555_{(21)}$ | $0.147_{(15)}$ | $0.489_{(11)}$ | $0.392_{(9)}$ | $0.720_{(6)}$ | 12.0 |
| FedFormer | $0.744_{(11)}$ | $0.317_{(13)}$ | $0.385_{(9)}$ | $0.341_{(10)}$ | $0.347_{(7)}$ | $0.233_{(21)}$ | $0.484_{(9)}$ | $0.413_{(13)}$ | $0.762_{(16)}$ | 12.1 |
| Autoformer | $0.751_{(12)}$ | $0.321_{(14)}$ | $0.392_{(12)}$ | $0.313_{(7)}$ | $0.354_{(9)}$ | $0.167_{(16)}$ | $0.484_{(9)}$ | $0.496_{(19)}$ | $0.756_{(13)}$ | 12.3 |
| Pyraformer | $0.781_{(18)}$ | $0.371_{(20)}$ | $0.390_{(10)}$ | $0.379_{(15)}$ | $0.385_{(15)}$ | $0.112_{(11)}$ | $0.493_{(13)}$ | $0.435_{(14)}$ | $0.735_{(9)}$ | 13.9 |
| Informer | $0.757_{(14)}$ | $0.336_{(17)}$ | $0.391_{(11)}$ | $0.383_{(16)}$ | $0.364_{(10)}$ | $0.192_{(18)}$ | $0.605_{(21)}$ | $0.542_{(20)}$ | $0.772_{(17)}$ | 16.0 |
| Transformer | $0.765_{(15)}$ | $0.321_{(14)}$ | $0.410_{(15)}$ | $0.405_{(18)}$ | $0.370_{(11)}$ | $0.178_{(17)}$ | $0.567_{(18)}$ | $0.592_{(21)}$ | $0.785_{(20)}$ | 16.6 |
| SCINet | $0.601_{(1)}$ | $0.193_{(1)}$ | $0.335_{(6)}$ | $0.280_{(4)}$ | $0.344_{(6)}$ | $0.137_{(13)}$ | $0.463_{(7)}$ | $0.389_{(7)}$ | $0.732_{(7)}$ | 5.8 |
| NLinear | $0.636_{(5)}$ | $0.223_{(5)}$ | $0.293_{(4)}$ | $0.239_{(1)}$ | $0.328_{(3)}$ | $0.091_{(7)}$ | $0.418_{(1)}$ | $0.375_{(4)}$ | $0.706_{(4)}$ | 3.8 |
| DLinear | $0.640_{(6)}$ | $0.497_{(23)}$ | $0.268_{(2)}$ | $0.336_{(9)}$ | $0.444_{(19)}$ | $0.102_{(10)}$ | $0.442_{(6)}$ | $0.378_{(5)}$ | $0.686_{(2)}$ | 9.1 |
| LSTMa | $0.974_{(23)}$ | $0.305_{(11)}$ | $0.510_{(18)}$ | $0.444_{(19)}$ | $0.501_{(20)}$ | $0.534_{(20)}$ | $0.782_{(23)}$ | $0.699_{(23)}$ | $0.897_{(23)}$ | 20.3 |

prediction horizon of $H = 168$ as in the previous experiment, we also use $H = 96, 192, 336$ and $720$ as in (Wu et al., 2021; Zhou et al., 2022b).

Table 3 shows the number of trainable parameters and inference time. As can be seen, `mr-Diff` has a smaller number of trainable parameters. Moreover, even with $S = 4$ or $5$ stages, its inference is more efficient than SSSD, CSDI, and TimeGrad. When $S = 3$ (the setting used in the previous experiments), `mr-Diff` is even faster than TimeDiff. The inference efficiency of `mr-Diff` over existing diffusion models is due to: (i) It is non-autoregressive, which avoids autoregressive decoding in each time step as in TimeGrad. (ii) It uses convolution layers. This avoids the scaling problem in CSDI, whose complexity is quadratic with the number of variables and time series length. (iii) Use of the acceleration technique DPM-Solver (Lu et al., 2022), which further reduces the number of denoising steps in each stage. Moreover, it is also faster than TimeDiff when $S = 2$ or $3$, thanks to its use of a linear mapping for future mixup instead of employing additional deep layers for this purpose.

Table 3: Number of trainable parameters and inference time (in ms) of various time series diffusion models with different prediction horizons ($H$) on the univariate *ETTh1*.

| | # of trainable params | inference time (ms) | | | | |
|---|---|---|---|---|---|---|
| | | $H = 96$ | $H = 168$ | $H = 192$ | $H = 336$ | $H = 720$ |
| `mr-Diff` ($S$=2) | **0.9M** | **8.3** | **9.5** | **9.8** | **11.9** | **21.6** |
| `mr-Diff` ($S$=3) | 1.4M | 12.5 | 14.3 | 14.9 | 16.8 | 27.5 |
| `mr-Diff` ($S$=4) | 1.8M | 16.7 | 19.1 | 19.7 | 28.5 | 36.4 |
| `mr-Diff` ($S$=5) | 2.3M | 30.0 | 30.2 | 30.2 | 35.0 | 43.6 |
| TimeDiff | 1.7M | 16.2 | 17.3 | 17.6 | 26.5 | 34.6 |
| TimeGrad | 3.1M | 870.2 | 1620.9 | 1854.5 | 3119.7 | 6724.1 |
| CSDI | 10M | 90.4 | 128.3 | 142.8 | 398.9 | 513.1 |
| SSSD | 32M | 418.6 | 590.2 | 645.4 | 1054.2 | 2516.9 |

## 6 CONCLUSION

In this paper, we propose the mutli-resolution diffusion model `mr-Diff`. Different from the existing time series diffusion model, `mr-Diff` incorporates seasonal-trend decomposition and uses multiple temporal resolutions in both the diffusion and denoising processes. By progressively denoising the time series in a coarse-to-fine manner, `mr-Diff` is able to produce more reliable predictions. Experiments on a number of real-world univariate and multivariate time series datasets show that `mr-Diff` outperforms the state-of-the-art time series diffusion models. It also achieves very competitive performance even when compared to various families of non-diffusion-based time series prediction models.

ACKNOWLEDGMENTS

This research was supported in part by the Research Grants Council of the Hong Kong Special Administrative Region (Grant 16200021), and the Science and Technology Planning Project of Guangdong Province (Grant 2023A0505050106).

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

## A  PSEUDOCODE

The training and inference procedures are shown in Algorithms 1 and 2, respectively.

---
**Algorithm 1** Training procedure of the backward denoising process.

---
1: **repeat**
2:    $(\mathbf{X}_0, \mathbf{Y}_0) \sim q(\mathbf{X}, \mathbf{Y})$
3:    $\{\mathbf{X}_s\}_{s=1,\ldots,S-1} = \texttt{TrendExtraction}(\mathbf{X}_0)$
4:    $\{\mathbf{Y}_s\}_{s=1,\ldots,S-1} = \texttt{TrendExtraction}(\mathbf{Y}_0)$
5:    **for** $s = S-1, \ldots, 0$ **do**
6:       $k \sim \texttt{Uniform}(\{1, 2, \ldots, K\})$
7:       $\epsilon \sim \mathcal{N}(\mathbf{0}, \mathbf{I})$
8:       Generate diffused sample $\mathbf{Y}_s^k$ by $\mathbf{Y}_s^k = \sqrt{\bar{\alpha}_k}\mathbf{Y}_s^0 + \sqrt{1 - \bar{\alpha}_k}\epsilon$
9:       Obtain diffusion step $k$'s embedding $\mathbf{p}^k$ using (7)
10:      Randomly generate a matrix $\mathbf{m}$ in (9)
11:      Obtain $\mathbf{z}_{\texttt{history}}$ using linear mapping on $\mathbf{X}_s$
12:      Obtain $\mathbf{z}_{\texttt{mix}}$ using (9)
13:      **if** $s < S-1$ **then**
14:         Obtain condition $\mathbf{c}_s$ by concatenating $\mathbf{z}_{\texttt{mix}}$ and $\mathbf{Y}_{s+1}^0$
15:      **else**
16:         Obtain condition $\mathbf{c}_s = \mathbf{z}_{\texttt{mix}}$
17:      **end if**
18:      Calculate the loss $\mathcal{L}_s^k(\theta_s)$ in (12)
19:      Take a gradient descent step based on $\nabla_{\theta_s}\mathcal{L}_k(\theta_s)$
20:    **end for**
21: **until** converged

---

---
**Algorithm 2** Inference procedure.

---
1: $\{\mathbf{X}_s\}_{s=1,\ldots,S-1} = \texttt{TrendExtraction}(\mathbf{X}_0)$
2: **for** $s = S-1, \ldots, 0$ **do**
3:    Obtain $\mathbf{z}_{\texttt{history}}$ using linear mapping on $\mathbf{X}_s$
4:    **if** $s < S-1$ **then**
5:       Obtain condition $\mathbf{c}_s$ by concatenating $\mathbf{z}_{\texttt{history}}$ and $\mathbf{Y}_{s+1}^0$
6:    **else**
7:       Obtain condition $\mathbf{c}_s = \mathbf{z}_{\texttt{history}}$
8:    **end if**
9:    Initialization: $\hat{\mathbf{Y}}_s^K \sim \mathcal{N}(\mathbf{0}, \mathbf{I})$
10:    **for** $k = K, \ldots, 1$ **do**
11:      $\epsilon \sim \mathcal{N}(\mathbf{0}, \mathbf{I})$, if $k > 1$, else $\epsilon = 0$
12:      Obtain diffusion step $k$'s embedding $\mathbf{p}^k$ using (7)
13:      Obtain the denoised output $\hat{\mathbf{Y}}_s^{k-1}$ by (13)
14:    **end for**
15: **end for**
16: **return** $\hat{\mathbf{Y}}_0^0$

---

## B  DATASETS

Experiments are performed on nine commonly-used real-world time series datasets (Zhou et al., 2021; Wu et al., 2021; Fan et al., 2022): (i) *NorPool* [1], which includes eight years of hourly energy production volume series in multiple European countries; (ii) *Caiso* [2], which contains eight years of hourly actual electricity load series in different zones of California; (iii) *Traffic* [3], which records the hourly road occupancy rates generated by sensors in the San Francisco Bay area freeways; (iv) *Electricity* [4], which includes the hourly electricity consumption of 321 clients over two years; (v) *Weather* [5], which records 21 meteorological indicators at 10-minute intervals from 2020 to 2021; (vi)

---
[1] https://www.nordpoolgroup.com/Market-data1/Power-system-data
[2] http://www.energyonline.com/Data
[3] http://pems.dot.ca.gov
[4] https://archive.ics.uci.edu/ml/datasets/ElectricityLoadDiagrams20112014
[5] https://www.bgc-jena.mpg.de/wetter/

*Exchange* [6] (Lai et al., 2018), which describes the daily exchange rates of eight countries (Australia, British, Canada, Switzerland, China, Japan, New Zealand, and Singapore); (vii)-(viii) *ETTh1* and *ETTm1*,[7] which contain two years of electricity transformer temperature data (Zhou et al., 2021) collected in China, at 1-hour and 15-minute intervals, respectively; (ix) *Wind* [8], which contains wind power records from 2020-2021 at 15-minute intervals (Li et al., 2022). Table 4 shows the dataset information. As different datasets have different sampling interval lengths, we follow (Shen & Kwok, 2023) and consider prediction tasks with more reasonable prediction lengths.

Table 4: Summary of dataset statistics, including the dimension, total number of observations, sampling frequency, and prediction length $H$ used in the experiments.

|  | dimension | #observations | frequency | steps ($H$) |
|---|---|---|---|---|
| *NorPool* | 18 | 70,128 | 1 hour | 720 (1 month) |
| *Caiso* | 10 | 74,472 | 1 hour | 720 (1 month) |
| *Traffic* | 862 | 17,544 | 1 hour | 168 (1 week) |
| *Electricity* | 321 | 26,304 | 1 hour | 168 (1 week) |
| *Weather* | 21 | 52,696 | 10 mins | 672 (1 week) |
| *Exchange* | 8 | 7,588 | 1 day | 14 (2 weeks) |
| *ETTh1* | 7 | 17,420 | 1 hour | 168 (1 week) |
| *ETTm1* | 7 | 69,680 | 15 mins | 192 (2 days) |
| *Wind* | 7 | 48,673 | 15 mins | 192 (2 days) |

Besides running directly on these multivariate datasets, we also convert them to univariate time series for performance comparison. For *NorPool* and *Caiso*, the univariate time series are extracted from all variables as in (Fan et al., 2022). For the other datasets, we follow (Wu et al., 2021; Zhou et al., 2021) and extract the univariate time series by using the last variable only.

Figure 5 shows examples of the time series data used in the experiments. Since all of them are multivariate, we only show the last variate. As can be seen, these datasets are representative in exhibiting various temporal characteristics (different stationary states, periodicities, and sampling rates). For example, *Electricity* contains obvious periodic patterns, while *ETTh1* and *ETTm1* exhibit non-stationary trends.

Real-world time series often have irregular dynamics over time or across variables. As in many deep time series models (Kim et al., 2021; Zhou et al., 2022a; Liu et al., 2022; Zeng et al., 2023), it is often useful to normalize the scales of the time series in each window. In the proposed method, we use instance normalization, which has also been used in RevIN (Kim et al., 2021) (which is referred to as reversible instance normalization) and FiLM (Zhou et al., 2022a). Specifically, we subtract the time series value in each window by its lookback window mean, and then divide by the lookback window's standard deviation. During inference, the mean and standard deviation are added back to the prediction.

## C  BASELINES IN MAIN EXPERIMENTS

We compare with several types of baselines, including: (i) recent time series diffusion models: non-autoregressive diffusion model TimeDiff (Shen & Kwok, 2023), TimeGrad (Rasul et al., 2021), conditional score-based diffusion model for imputation (CSDI) (Tashiro et al., 2021), structured state space model-based diffusion (SSSD) (Alcaraz & Strodthoff, 2022); (ii) recent generative models for time series prediction: variational autoencoder with diffusion, denoise and disentanglement (D$^3$VAE) (Li et al., 2022), coherent probabilistic forecasting (CPF) (Rangapuram et al., 2023), and PSA-GAN (Jeha et al., 2022); (iii) recent prediction models based on basis expansion: N-Hits (Challu et al., 2023), frequency improved Legendre memory model (FiLM) (Zhou et al., 2022a), Depts (Fan et al., 2022) and NBeats (Oreshkin et al., 2019); (iv) time series transformers: Scaleformer (Shabani et al., 2023), PatchTST (Nie et al., 2022), Fedformer (Zhou et al., 2022b), Autoformer (Wu et al., 2021), Pyraformer (Liu et al., 2021), Informer (Zhou et al., 2021) and the

---

[6]https://github.com/laiguokun/multivariate-time-series-data
[7]https://github.com/zhouhaoyi/ETDataset
[8]https://github.com/PaddlePaddle/PaddleSpatial/tree/main/paddlespatial/datasets/WindPower

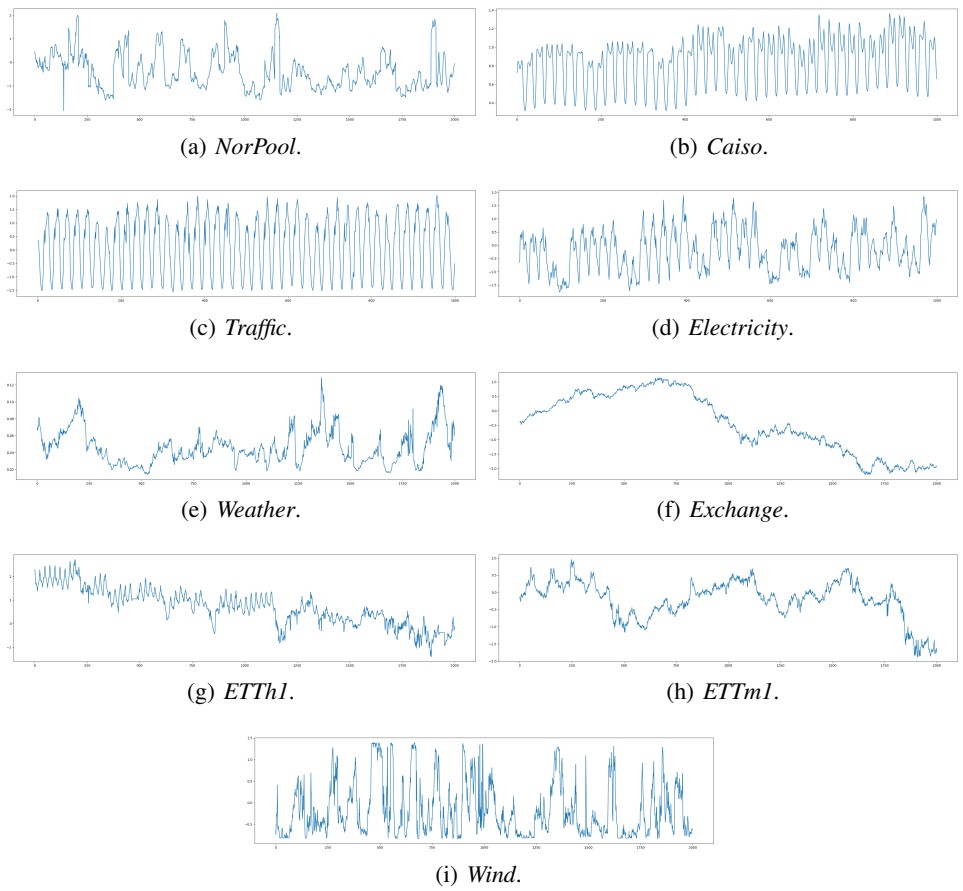

(a) *NorPool.*      (b) *Caiso.*

(c) *Traffic.*      (d) *Electricity.*

(e) *Weather.*      (f) *Exchange.*

(g) *ETTh1.*      (h) *ETTm1.*

(i) *Wind.*

Figure 5: Visualization of the time series datasets.

standard Transformer (Vaswani et al., 2017); and (v) other competitive baselines: SCINet (Liu et al., 2022) that introduces sample convolution and interaction for time series prediction, NLinear (Zeng et al., 2023), DLinear (Zeng et al., 2023) LSTMa (Bahdanau et al., 2015), and an attention-based LSTM (Hochreiter & Schmidhuber, 1997). We do not compare with (Yu et al., 2021; Niu et al., 2022; Madhusudhanan et al., 2021) because their codes are not publicly available.

## D    RESULTS ON MSE

Tables 5 and 6 show the MSE results on the univariate and multivariate time series, respectively, for the experiment in Section 5.1. As can be seen, the proposed mr-Diff still achieves the best overall MSE performance in terms of average rank. Moreover, since deep models for time series forecasting may be influenced by different random initializations, Table 7 shows the prediction results on the univariate time series over 5 random runs. As can be seen, the largest standard deviation is 0.0042, indicating that the proposed model is robust towards different initializations.

## E    ABLATION STUDIES

In this section, we perform a number of ablation experiments on mr-Diff using the *Electricity*, *ETTh1* and *ETTm1* datasets. These datasets are representative in exhibiting various temporal characteristics (different stationary states, periodicities, and sampling rates). For example, *Electricity* contains obvious periodic patterns, while *ETTh1* and *ETTm1* exhibit non-stationary trends.

Table 5: Univariate prediction MSEs on the real-world time series datasets (subscript is the rank). Results of all baselines (except NLinear, SCINet, and N-Hits) are from (Shen & Kwok, 2023).

| Method | NorPool | Caiso | Traffic | Electricity | Weather | Exchange | ETTh1 | ETTm1 | Wind | Rank |
|---|---|---|---|---|---|---|---|---|---|---|
| **mr-Diff** | $0.667_{(4)}$ | $0.122_{(3)}$ | $\mathbf{0.119}_{(1)}$ | $0.234_{(3)}$ | $\mathbf{0.002}_{(1)}$ | $\mathbf{0.016}_{(1)}$ | $\mathbf{0.066}_{(1)}$ | $0.039_{(2)}$ | $2.182_{(4)}$ | 2.2 |
| TimeDiff | $0.636_{(2)}$ | $0.122_{(3)}$ | $0.121_{(2)}$ | $0.232_{(2)}$ | $\mathbf{0.002}_{(1)}$ | $0.017_{(4)}$ | $\mathbf{0.066}_{(1)}$ | $0.040_{(5)}$ | $2.407_{(13)}$ | 3.7 |
| TimeGrad | $1.129_{(22)}$ | $0.325_{(22)}$ | $1.223_{(23)}$ | $0.920_{(23)}$ | $\mathbf{0.002}_{(1)}$ | $0.041_{(20)}$ | $0.078_{(11)}$ | $0.048_{(11)}$ | $2.530_{(18)}$ | 16.7 |
| CSDI | $0.967_{(21)}$ | $0.192_{(14)}$ | $0.393_{(20)}$ | $0.520_{(18)}$ | $\mathbf{0.002}_{(1)}$ | $0.071_{(23)}$ | $0.083_{(15)}$ | $0.050_{(15)}$ | $2.434_{(15)}$ | 15.8 |
| SSSD | $1.145_{(23)}$ | $0.176_{(12)}$ | $0.151_{(8)}$ | $0.370_{(11)}$ | $0.004_{(11)}$ | $0.023_{(16)}$ | $0.097_{(20)}$ | $0.049_{(13)}$ | $3.149_{(22)}$ | 15.1 |
| D³VAE | $0.964_{(20)}$ | $0.521_{(23)}$ | $0.151_{(8)}$ | $0.535_{(19)}$ | $0.003_{(9)}$ | $0.019_{(12)}$ | $0.078_{(10)}$ | $0.044_{(9)}$ | $2.679_{(20)}$ | 14.4 |
| CPF | $0.855_{(15)}$ | $0.260_{(21)}$ | $0.279_{(18)}$ | $0.609_{(21)}$ | $0.002_{(1)}$ | $0.016_{(1)}$ | $0.080_{(13)}$ | $0.041_{(6)}$ | $2.430_{(14)}$ | 12.2 |
| PSA-GAN | $0.658_{(3)}$ | $0.150_{(7)}$ | $0.250_{(17)}$ | $0.273_{(7)}$ | $0.035_{(21)}$ | $0.020_{(13)}$ | $0.084_{(16)}$ | $0.051_{(16)}$ | $2.510_{(17)}$ | 13.0 |
| N-Hits | $0.739_{(9)}$ | $0.170_{(11)}$ | $0.147_{(7)}$ | $0.346_{(8)}$ | $\mathbf{0.002}_{(1)}$ | $0.017_{(4)}$ | $0.089_{(17)}$ | $0.043_{(8)}$ | $2.406_{(12)}$ | 8.6 |
| FiLM | $0.707_{(7)}$ | $0.185_{(13)}$ | $0.198_{(13)}$ | $0.260_{(5)}$ | $0.007_{(14)}$ | $0.018_{(9)}$ | $0.070_{(3)}$ | $\mathbf{0.038}_{(1)}$ | $2.143_{(2)}$ | 7.4 |
| Depts | $0.668_{(5)}$ | $\mathbf{0.107}_{(1)}$ | $0.151_{(8)}$ | $0.380_{(14)}$ | $0.024_{(19)}$ | $0.020_{(13)}$ | $0.070_{(3)}$ | $0.046_{(10)}$ | $3.457_{(23)}$ | 10.7 |
| NBeats | $0.768_{(11)}$ | $0.125_{(5)}$ | $0.142_{(6)}$ | $0.378_{(13)}$ | $0.137_{(22)}$ | $\mathbf{0.016}_{(1)}$ | $0.095_{(19)}$ | $0.048_{(11)}$ | $2.434_{(15)}$ | 11.4 |
| Scaleformer | $0.778_{(12)}$ | $0.232_{(16)}$ | $0.286_{(19)}$ | $0.361_{(9)}$ | $0.009_{(17)}$ | $0.035_{(19)}$ | $0.150_{(22)}$ | $0.078_{(22)}$ | $2.646_{(19)}$ | 17.2 |
| PatchTST | $\mathbf{0.595}_{(1)}$ | $0.193_{(15)}$ | $0.177_{(12)}$ | $0.450_{(17)}$ | $0.026_{(20)}$ | $0.020_{(13)}$ | $0.106_{(21)}$ | $0.052_{(18)}$ | $2.698_{(21)}$ | 21 |
| FedFormer | $0.891_{(16)}$ | $0.164_{(9)}$ | $0.173_{(11)}$ | $0.376_{(12)}$ | $0.005_{(12)}$ | $0.050_{(22)}$ | $0.076_{(7)}$ | $0.065_{(21)}$ | $2.351_{(11)}$ | 13.4 |
| Autoformer | $0.946_{(19)}$ | $0.248_{(17)}$ | $0.473_{(21)}$ | $0.659_{(22)}$ | $0.003_{(9)}$ | $0.041_{(20)}$ | $0.081_{(14)}$ | $0.051_{(16)}$ | $2.349_{(10)}$ | 16.4 |
| Pyraformer | $0.933_{(18)}$ | $0.165_{(10)}$ | $0.136_{(4)}$ | $0.389_{(15)}$ | $0.020_{(18)}$ | $0.017_{(7)}$ | $0.076_{(7)}$ | $0.054_{(19)}$ | $2.279_{(6)}$ | 11.2 |
| Informer | $0.804_{(13)}$ | $0.250_{(18)}$ | $0.213_{(15)}$ | $0.363_{(10)}$ | $0.007_{(14)}$ | $0.023_{(16)}$ | $0.076_{(7)}$ | $0.049_{(13)}$ | $2.297_{(7)}$ | 12.6 |
| Transformer | $0.928_{(17)}$ | $0.250_{(18)}$ | $0.238_{(16)}$ | $0.430_{(14)}$ | $0.007_{(14)}$ | $0.018_{(9)}$ | $0.092_{(18)}$ | $0.058_{(20)}$ | $2.306_{(9)}$ | 15.2 |
| SCINet | $0.746_{(10)}$ | $0.154_{(8)}$ | $0.212_{(14)}$ | $0.272_{(6)}$ | $\mathbf{0.002}_{(1)}$ | $0.018_{(9)}$ | $0.071_{(6)}$ | $0.039_{(2)}$ | $\mathbf{2.063}_{(1)}$ | 6.3 |
| NLinear | $0.708_{(8)}$ | $0.147_{(6)}$ | $0.124_{(3)}$ | $\mathbf{0.231}_{(1)}$ | $\mathbf{0.002}_{(1)}$ | $0.017_{(4)}$ | $0.070_{(3)}$ | $0.039_{(2)}$ | $2.193_{(5)}$ | 3.7 |
| DLinear | $0.671_{(6)}$ | $0.118_{(2)}$ | $0.139_{(5)}$ | $0.244_{(4)}$ | $0.168_{(23)}$ | $0.017_{(4)}$ | $0.078_{(10)}$ | $0.041_{(6)}$ | $2.171_{(3)}$ | 7.0 |
| LSTMa | $0.836_{(14)}$ | $0.253_{(20)}$ | $1.032_{(22)}$ | $0.596_{(20)}$ | $0.005_{(12)}$ | $0.031_{(18)}$ | $0.167_{(23)}$ | $0.091_{(23)}$ | $2.299_{(8)}$ | 17.8 |

Table 6: Multivariate prediction MSEs on the real-world time series datasets (subscript is the rank). CSDI runs out of memory on *Traffic* and *Electricity*. Results of all baselines (except for NLinear, SCINet, and N-Hits) are from (Shen & Kwok, 2023).

| Method | NorPool | Caiso | Traffic | Electricity | Weather | Exchange | ETTh1 | ETTm1 | Wind | Rank |
|---|---|---|---|---|---|---|---|---|---|---|
| **mr-Diff** | $0.645_{(2)}$ | $0.127_{(3)}$ | $0.474_{(6)}$ | $0.155_{(3)}$ | $\mathbf{0.296}_{(1)}$ | $\mathbf{0.016}_{(1)}$ | $0.411_{(3)}$ | $0.340_{(2)}$ | $\mathbf{0.881}_{(1)}$ | 2.4 |
| TimeDiff | $0.665_{(4)}$ | $0.136_{(6)}$ | $0.564_{(7)}$ | $0.193_{(5)}$ | $0.311_{(2)}$ | $0.018_{(6)}$ | $\mathbf{0.407}_{(1)}$ | $\mathbf{0.336}_{(1)}$ | $0.896_{(2)}$ | 3.8 |
| TimeGrad | $1.152_{(20)}$ | $0.258_{(19)}$ | $1.745_{(22)}$ | $0.736_{(21)}$ | $0.392_{(14)}$ | $0.079_{(20)}$ | $0.993_{(22)}$ | $0.874_{(21)}$ | $1.209_{(21)}$ | 20.0 |
| CSDI | $1.011_{(19)}$ | $0.253_{(18)}$ | - | - | $0.356_{(9)}$ | $0.077_{(19)}$ | $0.497_{(7)}$ | $0.529_{(17)}$ | $1.066_{(9)}$ | 14.0 |
| SSSD | $0.872_{(12)}$ | $0.195_{(10)}$ | $0.642_{(11)}$ | $0.255_{(12)}$ | $0.349_{(8)}$ | $0.061_{(16)}$ | $0.726_{(18)}$ | $0.464_{(13)}$ | $1.188_{(19)}$ | 13.2 |
| D³VAE | $0.745_{(9)}$ | $0.241_{(17)}$ | $0.928_{(17)}$ | $0.286_{(15)}$ | $0.375_{(11)}$ | $0.200_{(22)}$ | $0.504_{(9)}$ | $0.362_{(8)}$ | $1.118_{(15)}$ | 13.7 |
| CPF | $1.613_{(23)}$ | $0.383_{(21)}$ | $1.625_{(21)}$ | $0.793_{(22)}$ | $1.390_{(23)}$ | $0.016_{(1)}$ | $0.730_{(19)}$ | $0.482_{(15)}$ | $1.140_{(17)}$ | 18.0 |
| PSA-GAN | $1.501_{(22)}$ | $0.510_{(23)}$ | $1.614_{(20)}$ | $0.535_{(20)}$ | $1.220_{(21)}$ | $0.018_{(6)}$ | $0.623_{(17)}$ | $0.537_{(18)}$ | $1.127_{(16)}$ | 18.1 |
| N-Hits | $0.716_{(7)}$ | $0.131_{(4)}$ | $0.386_{(2)}$ | $0.152_{(2)}$ | $0.323_{(4)}$ | $0.017_{(5)}$ | $0.498_{(8)}$ | $0.353_{(6)}$ | $1.033_{(6)}$ | 4.9 |
| FiLM | $0.723_{(8)}$ | $0.179_{(8)}$ | $0.628_{(10)}$ | $0.210_{(8)}$ | $0.327_{(5)}$ | $\mathbf{0.016}_{(1)}$ | $0.426_{(5)}$ | $0.347_{(4)}$ | $0.984_{(4)}$ | 5.9 |
| Depts | $0.662_{(3)}$ | $0.106_{(2)}$ | $1.019_{(19)}$ | $0.319_{(17)}$ | $0.761_{(19)}$ | $0.020_{(9)}$ | $0.579_{(13)}$ | $0.380_{(10)}$ | $1.082_{(12)}$ | 11.6 |
| NBeats | $0.832_{(10)}$ | $0.141_{(7)}$ | $\mathbf{0.373}_{(1)}$ | $0.269_{(13)}$ | $1.344_{(22)}$ | $\mathbf{0.016}_{(1)}$ | $0.586_{(15)}$ | $0.391_{(11)}$ | $1.069_{(10)}$ | 10.0 |
| Scaleformer | $0.983_{(15)}$ | $0.207_{(13)}$ | $0.618_{(9)}$ | $0.195_{(6)}$ | $0.462_{(16)}$ | $0.036_{(12)}$ | $0.613_{(16)}$ | $0.481_{(14)}$ | $1.359_{(22)}$ | 13.7 |
| PatchTST | $0.851_{(11)}$ | $0.193_{(9)}$ | $0.831_{(16)}$ | $0.225_{(10)}$ | $0.782_{(20)}$ | $0.047_{(14)}$ | $0.526_{(11)}$ | $0.372_{(9)}$ | $1.070_{(11)}$ | 12.3 |
| FedFormer | $0.873_{(13)}$ | $0.205_{(11)}$ | $0.591_{(8)}$ | $0.238_{(11)}$ | $0.342_{(7)}$ | $0.133_{(21)}$ | $0.541_{(12)}$ | $0.426_{(12)}$ | $1.113_{(14)}$ | 12.1 |
| Autoformer | $0.940_{(14)}$ | $0.226_{(15)}$ | $0.688_{(15)}$ | $0.201_{(7)}$ | $0.360_{(10)}$ | $0.056_{(15)}$ | $0.516_{(10)}$ | $0.565_{(19)}$ | $1.083_{(13)}$ | 13.1 |
| Pyraformer | $1.008_{(18)}$ | $0.273_{(20)}$ | $0.659_{(12)}$ | $0.273_{(14)}$ | $0.394_{(15)}$ | $0.032_{(11)}$ | $0.579_{(13)}$ | $0.493_{(16)}$ | $1.061_{(8)}$ | 14.1 |
| Informer | $0.985_{(16)}$ | $0.231_{(16)}$ | $0.664_{(13)}$ | $0.298_{(16)}$ | $0.385_{(12)}$ | $0.073_{(18)}$ | $0.775_{(21)}$ | $0.673_{(20)}$ | $1.168_{(18)}$ | 16.7 |
| Transformer | $1.005_{(17)}$ | $0.206_{(12)}$ | $0.671_{(14)}$ | $0.328_{(18)}$ | $0.388_{(13)}$ | $0.062_{(17)}$ | $0.759_{(20)}$ | $0.992_{(22)}$ | $1.201_{(20)}$ | 17.0 |
| SCINet | $\mathbf{0.613}_{(1)}$ | $\mathbf{0.095}_{(1)}$ | $0.434_{(5)}$ | $0.171_{(4)}$ | $0.329_{(6)}$ | $0.036_{(12)}$ | $0.465_{(6)}$ | $0.359_{(7)}$ | $1.055_{(7)}$ | 5.4 |
| NLinear | $0.707_{(6)}$ | $0.135_{(5)}$ | $0.430_{(4)}$ | $\mathbf{0.147}_{(1)}$ | $0.313_{(3)}$ | $0.019_{(8)}$ | $0.410_{(2)}$ | $0.349_{(5)}$ | $0.989_{(5)}$ | 4.3 |
| DLinear | $0.670_{(5)}$ | $0.461_{(22)}$ | $0.389_{(3)}$ | $0.215_{(9)}$ | $0.488_{(17)}$ | $0.022_{(10)}$ | $0.415_{(4)}$ | $0.345_{(3)}$ | $0.899_{(3)}$ | 8.4 |
| LSTMa | $1.481_{(21)}$ | $0.217_{(14)}$ | $0.966_{(18)}$ | $0.414_{(19)}$ | $0.662_{(18)}$ | $0.403_{(23)}$ | $1.149_{(23)}$ | $1.030_{(23)}$ | $1.464_{(23)}$ | 20.2 |

**Length of lookback window** $L$. Table 8 shows the prediction MAE of `mr-Diff` with different lengths ($L$) of the lookback window. We consider $L = \{96, 192, 336, 720, 1440\}$ as used in (Shen & Kwok, 2023; Zeng et al., 2023; Wu et al., 2021). As can be seen, on *Electricity*, *ETTh1* and

Table 7: Univariate prediction errors of mr-Diff obtained on five runs.

| | NorPool | | Caiso | | Traffic | | Electricity | |
|---|---|---|---|---|---|---|---|---|
| | MAE | MSE | MAE | MSE | MAE | MSE | MAE | MSE |
| 0 | 0.6096 | 0.6668 | 0.2129 | 0.1225 | 0.1973 | 0.1192 | 0.3322 | 0.2331 |
| 1 | 0.6092 | 0.6664 | 0.2131 | 0.1226 | 0.1974 | 0.1195 | 0.3267 | 0.2293 |
| 2 | 0.6087 | 0.6659 | 0.2091 | 0.1220 | 0.1970 | 0.1193 | 0.3327 | 0.2409 |
| 3 | 0.6099 | 0.6672 | 0.2129 | 0.125 | 0.1976 | 0.1192 | 0.3382 | 0.2347 |
| 4 | 0.6091 | 0.6665 | 0.2120 | 0.1221 | 0.1973 | 0.1195 | 0.3321 | 0.2339 |
| mean | 0.6093 | 0.6666 | 0.2120 | 0.1223 | 0.1973 | 0.1193 | 0.3324 | 0.2343 |
| std deviation | 0.0005 | 0.0005 | 0.0017 | 0.0003 | 0.0002 | 0.0001 | 0.0041 | 0.0042 |

| | Weather | | Exchange | | ETTh1 | | ETTm1 | | Wind | |
|---|---|---|---|---|---|---|---|---|---|---|
| | MAE | MSE | MAE | MSE | MAE | MSE | MAE | MSE | MAE | MSE |
| 0 | 0.0322 | 0.0019 | 0.0936 | 0.0155 | 0.1964 | 0.0663 | 0.1486 | 0.0389 | 1.1704 | 2.1862 |
| 1 | 0.0321 | 0.0018 | 0.0935 | 0.0155 | 0.1968 | 0.0667 | 0.1489 | 0.0387 | 1.1682 | 2.1820 |
| 2 | 0.0323 | 0.0018 | 0.0947 | 0.0158 | 0.1956 | 0.0660 | 0.1483 | 0.0384 | 1.1690 | 2.1822 |
| 3 | 0.0322 | 0.0019 | 0.0954 | 0.0160 | 0.1959 | 0.0661 | 0.1486 | 0.0387 | 1.1676 | 2.1807 |
| 4 | 0.0322 | 0.0018 | 0.0946 | 0.0159 | 0.1965 | 0.0671 | 0.1489 | 0.0389 | 1.1670 | 2.1804 |
| mean | 0.0322 | 0.0018 | 0.0943 | 0.0157 | 0.1962 | 0.0664 | 0.1486 | 0.0387 | 1.1684 | 2.1823 |
| std deviation | 0.0001 | 0.0001 | 0.0008 | 0.0002 | 0.0005 | 0.0004 | 0.0003 | 0.0002 | 0.0013 | 0.0023 |

*ETTm1* (corresponding to 168/168/192-step-ahead prediction, respectively), good performance can be obtained when $L$ is 336 or above.

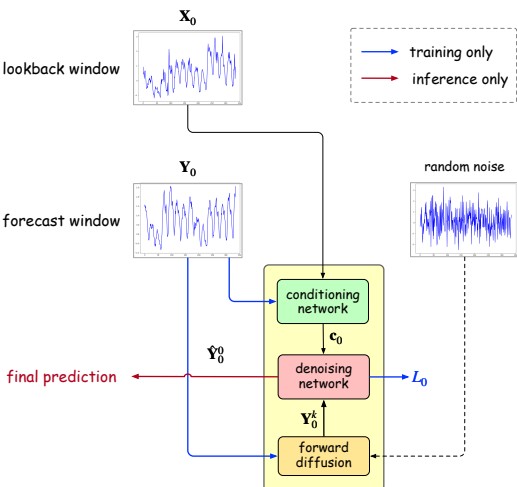

Figure 6: The baseline without seasonal-trend decomposition blocks.

Table 8: Prediction MAE versus lookback window length $L$.

| $L$ | Electricity | ETTh1 | ETTm1 |
|---|---|---|---|
| 96 | 0.449 | 0.210 | 0.165 |
| 192 | 0.376 | 0.202 | 0.163 |
| 336 | 0.362 | **0.196** | 0.157 |
| 720 | **0.332** | 0.196 | 0.152 |
| 1,440 | 0.346 | 0.199 | **0.149** |

**Number of stages $S$.** In this experiment, we vary the number of stages $S$. The kernel size $\tau_s$ is set to increase with the stage number $s$, so as to generate fine-to-coarse trends. We also compare with a variant that does not perform seasonal-trend decomposition (illustrated in Figure 6). This variant directly generates the forecast window from a random noise vector without using the coarser

trend $\mathbf{Y}_{s+1}^0$. Table 9 shows the prediction errors. As can be seen, using multiple stages improves performance, and not using the seasonal-trend decomposition leads to the worst performance.

Table 9: Prediction MAE versus number of stages $S$. $L$ is set to the optimal setting in Table 8 (i.e., *Electricity*: 720, *ETTh1*: 336, *ETTm1*: 1440.

| $S$ | *Electricity* | *ETTh1* | *ETTm1* |
|-----|---------------|---------|---------|
| 1   | 0.403         | 0.208   | 0.1573  |
| 2   | 0.389         | 0.200   | 0.1529  |
| 3   | 0.363         | **0.196** | 0.1525  |
| 4   | 0.346         | 0.197   | 0.1509  |
| 5   | **0.332**     | 0.197   | **0.1496** |

**Number of diffusion steps** $K$. Table 10 shows the prediction error of `mr-Diff` with $K$, the number of diffusion steps. As can be seen, setting $K = 100$ leads to stable performance across all four datasets. This also agrees with (Li et al., 2022) in that a small $K$ may lead to incomplete diffusion, while a $K$ too large may involve unncessary computations.

Table 10: Prediction MAE versus number of diffusion steps $K$.

| $K$   | *Electricity* | *ETTh1* | *ETTm1* |
|-------|---------------|---------|---------|
| 50    | 0.348         | 0.199   | 0.151   |
| 100   | **0.332**     | **0.196** | **0.149** |
| 500   | 0.335         | 0.198   | 0.151   |
| 1,000 | 0.337         | 0.199   | 0.151   |

**Gaussian noise variance** $\beta_k$. Recall from Section 5 that $\beta_k$ is generated by a linear variance schedule (Rasul et al., 2021). In this experiment, we vary $\beta_K$ in {0.001,0.01,0.1,0.9}, with $\beta_1$ fixed to $10^{-4}$. As can be seen from Table 11, setting $\beta_K = 0.1$ always has the best performance. Li et al. (2022) has a similar conclusion that a $\beta_K$ too small can lead to a unsatisfactory diffusion, while a $\beta_K$ too large can make the diffusion out of control.

Table 11: Prediction MAE versus Gaussian noise variance upper bound $\beta_k$.

| $\beta_K$ | *Electricity* | *ETTh1* | *ETTm1* |
|-----------|---------------|---------|---------|
| 0.001     | 0.403         | 0.198   | 0.151   |
| 0.01      | 0.365         | 0.198   | 0.152   |
| 0.1       | **0.332**     | **0.196** | **0.149** |
| 0.9       | 0.372         | 0.201   | 0.154   |

**Future mixup.** Recall that each element of the future mixup matrix $\mathbf{m}$ in (9) is randomly sampled from the uniform distribution on $[0; 1)$. In this experiment, we study the effect of $\mathbf{m}$ by sampling each of its elements from the Beta distribution $Beta(\gamma, \gamma)$ where $\gamma \in \{0.1, 1, 2\}$. Note that $Beta(1, 1)$ reduces to the uniform distribution. Also, we include a `mr-Diff` variant that does not use future mixup (by replacing (9) with $\mathbf{z_{mix}} = \mathbf{z}_{history}$). Table 12 shows the prediction results. As can be seen, the uniform distribution (with $\gamma = 1$) as used in `mr-Diff` has the best performance.

## F MORE IMPLEMENTATION DETAILS

In the proposed `mr-Diff`, the conditioning network and the denoising network's encoder/decoder are built by stacking a number of convolutional blocks. The default configuration of each convolutional block is shown in Table 13.

The number $S$ is selected from $\{2, 3, 4, 5\}$ (the number of stages $S$ is greater than the number of decompositions by 1). When $S = 2$, the kernel size $\tau_1$ is selected from $\{5, 25\}$; When $S = 3$,

Table 12: Prediction MAE with future mixup matrix **m** from different Beta distributions $Beta(\gamma, \gamma)$.

| $\gamma$ | Electricity | ETTh1 | ETTm1 |
|---|---|---|---|
| 0.1 | 0.348 | 0.202 | 0.154 |
| 1 | **0.332** | **0.196** | **0.149** |
| 2 | 0.776 | 0.222 | 0.200 |
| variant without future mixup | 0.349 | 0.198 | 0.151 |

Table 13: Configuration of the convolutional block.

| layer | operator | default parameters |
|---|---|---|
| 1 | Conv1d | in channel=256, out channel=256, kernel size=3, stride=1, padding=1 |
| 2 | BatchNorm1d | number of features=256 |
| 3 | LeakyReLU | negative slope=0.1 |
| 4 | Dropout | dropout rate=0.1 |

the kernel size $(\tau_1, \tau_2)$ is selected from $\{(5, 25), (25, 51), (51, 201)\}$; When $S = 4$, the kernel size $(\tau_1, \tau_2, \tau_3)$ is selected from $\{(5, 25, 51), (25, 51, 201)\}$. When $S = 5$, the kernel size $(\tau_1, \tau_2, \tau_3, \tau_4)$ is selected from $\{(5, 25, 51, 201)\}$. In practice, we run a grid search over $S$ for each dataset. This is not expensive as the number of choices is small (as can be seen from above). Usually, $S = 2$ and the combinations (5,25) and (25,51) achieve promising performance.

## G  TRAINING EFFICIENCY

In Section 5.2, we discussed inference efficiency. In this section, we further compare training efficiency of mr-Diff with the other time series diffusion model baselines (TimeDiff, TimeGrad, CSDI, SSSD) on the univariate *ETTh1* dataset. As can be seen from Table 14, the proposed model requires less training time compared to TimeGrad, CSDI, and SSSD. This suggests that the proposed model is more efficient in terms of training. This is because, during training, mr-Diff uses convolution layers in its denoising networks, which eliminates the need for large modules like residual blocks (in TimeGrad and SSSD) and self-attention layers (in CSDI). Moreover, it is also faster than TimeDiff, thanks to its use of a linear mapping for future mixup instead of employing additional deep layers for this purpose.

Table 14: Training time (ms) of various time series diffusion models with different prediction horizons ($H$) on the univariate *ETTh1*.

| | training time (ms) | | | | |
|---|---|---|---|---|---|
| | $H = 96$ | $H = 168$ | $H = 192$ | $H = 336$ | $H = 720$ |
| mr-Diff ($S$=2) | **0.54** | **0.57** | **0.58** | **0.62** | **0.66** |
| mr-Diff ($S$=3) | 0.59 | 0.69 | 0.71 | 0.74 | 0.82 |
| mr-Diff ($S$=4) | 0.78 | 0.96 | 0.99 | 1.07 | 1.45 |
| mr-Diff ($S$=5) | 1.19 | 1.26 | 1.27 | 1.31 | 1.72 |
| TimeDiff | 0.71 | 0.75 | 0.77 | 0.82 | 0.85 |
| TimeGrad | 2.11 | 2.42 | 3.21 | 4.22 | 5.93 |
| CSDI | 5.72 | 7.09 | 7.59 | 10.59 | 17.21 |
| SSSD | 16.98 | 19.34 | 22.64 | 32.12 | 52.93 |

