# OpenReview forum: "Multi-Resolution Diffusion Models for Time Series Forecasting"
_ICLR.cc/2024/Conference — ICLR 2024 poster_

### Official Review · Reviewer_WNFV · 2023-10-23

**Soundness:** 3 good
**Presentation:** 3 good
**Contribution:** 2 fair
**Rating:** 6
**Confidence:** 3

**Summary:**

In this paper, the authors proposed a new diffusion-based time-series forecasting model that involves a coarse-to-fine trend generation process.  The empirical study shows that the proposed model outperforms the state-of-the-art diffusion models proposed in this area and has the comparable performance in comparison to the best existing time-series forecasting models.

**Strengths:**

1. The proposed model achieves the state-of-the-art among the diffusion-based forecasting models in terms of the inference speed and the output quality. Additionally, according to Table 1, it also has the best performance on average among the listed forecasting models.

**Weaknesses:**

1. It appears that the networks at different stages are independent, which suggests that the total number of parameters in the proposed model is significantly higher than the ones in other models. If so, the performance of mr-diff would be less impressive than what appears in the paper. I think the authors should discuss this issue in the paper to avoid potential misinterpretation.
2. More details should be given regarding the seasonal-trend decomposition and the generation of the coarse trends.

**Questions:**

Please check the Weaknesses section.

Additionally, the paper claims that the proposed model enjoys a lower computational complexity by dropping the attention layer in their architecture; however, this may result that when the generated time series is sufficiently long, the beginning and the end of the series might be independent of each other and cause some inconsistency problem. I wonder if the authors could give some comments on this issue.



---- post rebuttal ----

I thank the authors for the clarifications and the additional results. All my concerns have been addressed, and I have adjusted the score correspondingly. I hope the authors can include the results in the final version if the paper gets accepted.

---

> ### Author Response · Authors · 2023-11-21
>
> **Q1: It appears that the networks at different stages are independent, which suggests that the total number of parameters in the proposed model is significantly higher than the ones in other models. If so, the performance of mr-diff would be less impressive than what appears in the paper. I think the authors should discuss this issue in the paper to avoid potential misinterpretation.**
>
> **A1:** Indeed, the number of parameters in mr-Diff is often smaller than that in other diffusion models.
> The table below compares the number of trainable parameters for the various time series diffusion models.
> As can be seen, the number of parameters in mr-Diff is smaller than those in TimeGrad, CSDI and SSSD. This is because mr-Diff uses convolution layers in its denoising networks, which eliminates the need for
> large modules like residual blocks (TimeGrad and SSSD) or self-attention layers (CSDI).
>
> |      | \# of trainable params |
> | ---- | ---------------------- |
> |mr-Diff (S=2) |  0.9M |
> |mr-Diff (S=3) |  1.4M |
> |mr-Diff (S=4) |  1.8M |
> |mr-Diff (S=5) |  2.3M |
> |TimeDiff |  1.7M |
> |TimeGrad |  3.1M |
> |CSDI |  10M |
> |SSSD |  32M |
>
> **Q2: More details should be given regarding the seasonal-trend decomposition and the generation
> of the coarse trends.**
>
> **A2:** The seasonal-trend decomposition is a classic time series method that aims to extract informative trend and seasonal components from time series [1]. The decomposition method utilized in this paper is described by the trend extraction equation presented in Section 4.1, which corresponds to the decomposition method described in [2]. At each decomposition stage, it takes a finer trend (such as $\mathbf{X}_1$ and $\mathbf{Y}_1$ in Figure 2) as the input and generates a coarser trend (such as $\mathbf{X}_2$ and $\mathbf{Y}_2$ in Figure 2).
>
> **References:**
>
> - [1] Cleveland R B, Cleveland W S, McRae J E, et al. STL: A seasonal-trend decomposition. J. Off. Stat, 1990, 6(1): 3-73.
>
> - [2] NeurIPS'21 Autoformer: Decomposition transformers with auto-correlation for long-term series forecasting.
>
> **Q3: Additionally, the paper claims that the proposed model enjoys a lower computational complexity by dropping the attention layer in their architecture; however, this may result that when the generated time series is sufficiently long, the beginning and the end of the series might be independent of each other and cause some inconsistency problem. I wonder if the authors could give some comments on this issue.**
>
> **A3:** This problem is not observed in our long-term forecasting experiments (e.g., with prediction horizon $H = 720$ on NorPool and Caiso and $H = 168$ for Traffic and Electricity). These $H$ values are usually considered long-term forecasting in the literature. As can be seen from Tables 1-2 and 13-14, mr-Diff consistently achieves the highest average ranking. This demonstrates the effectiveness of mr-Diff for long-term forecasting. This is mainly because the proposed **mr-Diff combines the merits of diffusion model and multiresolution analysis**. The diffusion model provides a strong ability for data generation and **multiresolution analysis allows the use of coarser trends to help capture long-term dependencies**.

---

### Official Review · Reviewer_xEhe · 2023-10-31

**Soundness:** 3 good
**Presentation:** 3 good
**Contribution:** 3 good
**Rating:** 6
**Confidence:** 4

**Summary:**

In this paper, the authors proposed a diffusion model for the time series forecasting problem. Specifically, they introduced a solution for applying multi-resolution analysis in a time series diffusion model.

**Strengths:**

1. The problem studied in this paper is interesting and important
2. The proposed solution enables the diffusion model to handle and utilize multi-resolution time series.

**Weaknesses:**

1. More analysis is needed on the experiment results.

**Questions:**

The authors proposed a neat solution to make use of the multi-resolution analysis in a diffusion model for the time series forecasting. I have following questions regarding the experiment results:
1. The experiment results seem suggest that the performance does not benefit a lot from the multi-resolution analysis. The performance gain is not significant after comparing it with the best models in diffusion models, basis expansion-based models, and the last category of models. The authors claimed that the lack of complex trend information is responsible for the absence of improvement. Could the authors provide further elaboration on this point? What could be considered as complex trends that the proposed model can take advantage of?
2. What is the inference time of the proposed model compared to NLinear and DLinear?
3. The performance gain in multivariate prediction is even less significant when comparing the data in Table 1 and 2, as well as in Table 13 and 14. Can the authors elaborate on this observation?

---

> ### Author Response · Authors · 2023-11-21
>
> **Q1: More analysis is needed on the experiment results. The experiment results seem to suggest that the performance does not benefit a lot from the multi-resolution analysis. The performance gain is not significant after comparing it with the best models in diffusion models, basis expansion-based models, and the last category of models. The authors claimed that the lack of complex trend information is responsible for the absence of improvement. Could the authors provide further elaboration on this point? What could be considered as complex trends that the proposed model can take advantage of?**
>
> **A1:** We respectfully disagree with the statement that "The performance gain is not significant after comparing it with the best models in diffusion models, basis expansion-based models, and the last category of models''. It is difficult for a method to outperform all baselines on all datasets, and the ranking of mr-Diff is higher than all other baselines.
>
> Specifically, in Table 1, mr-Diff is the best in 5 of the 9 datasets and the second best in 3 datasets. In comparison, the best diffusion model baseline is TimeDiff, which ranks second best on only one dataset. The best expansion-based model, FiLM, achieves the best on only one dataset. The best model in the last category, NLinear, is the best on two datasets and the second best in two datasets.
>
> We mentioned in Section 5.1 that  *there is no improvement on datasets such as Caiso. This is because there is no complex trend information on this dataset that can be leveraged (as shown in Figure 5 (b) in Appendix B)*. As can be seen from Figure 5 (b), the Caiso data is relatively simple as it presents clear periodicity. Hence, on this easy dataset, the improvement is not very significant. However, the dataset ETTh1 (Figure 4) is more challenging as it exhibits complex temporal behaviors (such as nonstationarity, nonlinear temporal patterns, irregular oscillations, and abrupt changes) that are not easily captured by simple or linear trend models. By combining the merits of diffusion models and multiresolution analysis, the proposed mr-Diff achieves better results in this data.
>
>
>
> **Q2: What is the inference time of the proposed model compared to NLinear and DLinear?**
>
> **A2:** As suggested, we include the inference time (in ms) of NLinear and DLinear at different prediction horizons ($H$) in the table below. Note that NLinear and DLinear are linear models, and thus have faster training and inference compared to the proposed model and other deep learning methods. However, their performance are limited on the more complicated datasets. For example, as can be seen from Table 1,
> NLinear ranks 6 and 4 on the NorPool and ETTh1 datasets, respectively; whereas the proposed mr-Diff ranks 2 on both datasets. Compared to the other time series diffusion models, it is important to note that
> the proposed mr-Diff model is much faster than TimeGrad, CSDI and SSSD, and is faster or comparable
> with TimeDiff depending on the setting of $S$.
>
> | |$H=96$ |  $H=168$ |  $H=192$ | $H=336$ |  $H=720$  |
> | ---- | ---- | ---- | ---- | ---- | ---- |
> | mr-Diff ($S$=2) | 8.3 |  9.5    | 9.8 |  11.9 |   21.6 |
> | mr-Diff ($S$=3) | 12.5 |  14.3  | 14.9 |  16.8  |  27.5|
> | mr-Diff ($S$=4) | 16.7 |    19.1 | 19.7 |     28.5 |   36.4|
> | mr-Diff ($S$=5) | 30.0 |  30.2 | 30.2 |   35.0 |   43.6 |
> | TimeDiff |  16.2 |  17.3 |  17.6 |  26.5 |  34.6 |
> | TimeGrad |  870.2|  1620.9 |  1854.5 |  3119.7 |  6724.1 |
> | CSDI |  90.4 |  128.3 |  142.8 |  398.9 |  513.1|
> | SSSD |  418.6 |  590.2  |  645.4 |  1054.2 |  2516.9 |
> | **NLinear** |  0.01 |  0.01 |  0.01 |  0.01 |  0.01|
> | **DLinear** |  0.02 |  0.02 |  0.02 |  0.02 |  0.02|
>
> **Q3: The performance gain in multivariate prediction is even less significant when comparing the data in Tables 1 and 2, as well as in Tables 13 and 14. Can the authors elaborate on this observation?**
>
> **A3:**  We respectfully disagree with that "The performance gain in multivariate prediction is even less significant". **The results in Tables 1-2 and 13-14 show that the proposed mr-Diff consistently ranks higher than the other baselines across all nine datasets**. In particular, for multivariate prediction, it is best on 1 dataset and second-best on 4 of 9 datasets (Table 2). In Table 14, its MSE is the best on 3 of 9 datasets and second-best on 2 datasets.

---

### Official Review · Reviewer_V8ZH · 2023-11-05

**Soundness:** 3 good
**Presentation:** 3 good
**Contribution:** 3 good
**Rating:** 8
**Confidence:** 3

**Summary:**

This paper proposes and studies a novel diffusion model for forecasting time series. The model utilizes the seasonal-trend decomposition to sequentially extract fine-to-coarse trends from the time series for forward diffusion, and the denoising process proceeds by generating the coarsest trend first and the finer details are progressively added (using the predicted coarser trends as condition variables).

**Strengths:**

- Overall the paper is well written
- A solid contribution to extend recent diffusion models for time series forecasting by leveraging a seasonal-trend decomposition

**Weaknesses:**

- The proposed model is restricted to/evaluated on only forecasting tasks but not on generation tasks and downstream tasks like classification (c.f. https://arxiv.org/abs/2303.09489)
- The proposed model is somewhat complicated, consisting of many components, and it is not clear how the model performance would be affected if one of the components is not used, as well as how this depends on the multiscale nature of the time series. A systematic approach is to carefully evaluating the model on synthetic time series data generated by multiscale dynamical systems; e.g., multiscale Lorenz-96 systems
- Missing discussions on training efficiency and number of trainable parameters, and how do they compare to other models
- Missing evaluations on other forecasting tasks in the Monash benchmark:
https://forecastingdata.org/
- Error bars evaluation is not completed (provided for only Electricity, ETTh1, ETTm1 but not the others)

**Questions:**

- MAE and MSE are used as metrics to evaluate the forecasting performance. How about using MAPE?
- How effective is the model for long-term forecasting (i.e. when H is large)? Can it reconstruct the long term statistics of the underlying dynamical system accurately?
- What are the limitations of the proposed model? How will the proposed model perform on all the tasks of the Monash benchmark?

---

> ### Author Response · Authors · 2023-11-21
>
> **Q1: The proposed model is restricted to/evaluated on only forecasting tasks but not on generation tasks and downstream tasks like classification (c.f. https://arxiv.org/abs/2303.09489)**
>
> **A1:** This paper focuses on time series forecasting by incorporating multiresolution analysis into time series diffusion models. A number of popular related papers [1-5] also focus only on time series forecasting.  Moreover, as mentioned in Section 3.2, time series forecasting requires predicting future values conditioned on past observations, and thus forecasting can also be regarded as generation. Extending the proposed diffusion model to classification is an interesting topic, and will be studied in the future.
>
> **References:**
>
> - [1] ICML'23 Non-autoregressive conditional diffusion models for time series prediction.
>
> - [2] ICLR'23 Scaleformer: Iterative multi-scale refining transformers for time series forecasting.
>
> - [3] ICLR'22 DEPTS: Deep expansion learning for periodic time series forecasting.
>
> - [4] NeurIPS'22 Generative time series forecasting with diffusion, denoise, and disentanglement.
>
> - [5] ICML'22 FEDformer: frequency enhanced decomposed transformer for long-term series forecasting.
>
> **Q2:  The proposed model is somewhat complicated, consisting of many components, and it is not clear how the model performance would be affected if one of the components is not used, as well as how this depends on the multiscale nature of the time series. A systematic approach is to carefully evaluate the model on synthetic time series data generated by multiscale dynamical systems; e.g., multiscale Lorenz-96 systems**
>
> **A2:** There might be some misunderstanding. Indeed, as mentioned in the first paragraph of Section 5, we have provided further studies on hyperparameters/components in the proposed mr-Diff. However, because of the lack of space, these ablation studies are **reported in Appendix F**. In particular, we **have studied three important components**: i) replacing the seasonal-trend decomposition by downsampling/upsampling operations, ii) varying the number of decomposition stages $S$ (note that using $S=1$ is the same as not performing seasonal-trend decomposition), and iii) removing future mixup. The results of these studies demonstrate the effectiveness of these components in the proposed diffusion model.
>
> As suggested by the reviewer, we evaluate the proposed model on the multiscale Lorenz-96 systems. The prediction length $H$ is set to 96. The MAE results are shown in the table below. As can be seen, mr-Diff consistently outperforms other baselines, as it combines the merits of diffusion models and multiresolution analysis.
>
> |mr-Diff  |  TimeDiff |  TimeGrad |  N-Hits |  FiLM |   PatchTST  |    Fedformer |  SCINet |     NLinear |
> | ---- | ---------------------- | ---- | ----- | ----- | ----- | ----- |----- | ----- |
> |$\mathbf{0.730}$ |  0.789 |  0.970 |  0.822 |  $\underline{0.761}$ |  0.879 |  0.795 |  0.843 |  0.788 |
>
>
> **Q3:  Missing discussions on training efficiency and number of trainable parameters, and how do they compare to other models**
>
> **A3:** As suggested by the reviewer, the table below shows the number of trainable parameters and the training time (in ms) of different time series diffusion models. As can be seen, the number of trainable parameters is smaller than those of TimeGrad, CSDI and SSSD. This is because mr-Diff uses convolution layers in its denoising networks, which eliminates the need for large modules like residual blocks (in TimeGrad and SSSD) and self-attention layers (iin CSDI). Moreover, mr-Diff is also faster than TimeDiff, thanks to its use of a linear mapping for future mixup instead of employing additional deep layers for this purpose.
>
> |      | \# of trainable params | H=96 | H=168 | H=192 | H=336 | H=720 |
> | ---- | :--------------------: | :----: | :-----: | :-----: | :-----: | :-----: |
> | mr-Diff (S=2) | 0.9M | 0.54 | 0.57 | 0.58 | 0.62 | 0.66 |
> | mr-Diff (S=3) | 1.4M | 0.59 | 0.69 | 0.71 | 0.74 | 0.82 |
> | mr-Diff (S=4) | 1.8M |  0.78  |0.96   |0.99   | 1.07   | 1.45 |
> | mr-Diff (S=5) |  2.3M |  1.19     | 1.26   | 1.27   | 1.31   | 1.72 |
> | TimeDiff |  1.7M |  0.71  | 0.75   | 0.77   | 0.82   | 0.85 |
> | TimeGrad |  3.1M |  2.11  | 2.42   | 3.21   | 4.22   | 5.93 |
> | CSDI |  10M |  5.72   | 7.09   | 7.59   | 10.59  | 17.21 |
> | SSSD |  32M |  16.98  | 19.34  | 22.64  | 32.12  | 52.93 |

---

> ### Author Response · Authors · 2023-11-21
>
> **Q4: Missing evaluations on other forecasting tasks in the Monash benchmark. How will the proposed model perform on all the tasks of the Monash benchmark?**
>
> **A4:** As suggested by the reviewer, we add experiments on the Monash benchmark. Because of the limited time of the rebuttal period, only 12 Monash datasets are used. Note that the submission already includes nine real-world datasets that exhibit diverse temporal characteristics (such as different stationary states, periodicities, and sampling rates) and have been commonly used in related time series prediction works (including Autoformer (NeurIPS'21), SCINet (NeurIPS'22), FiLM (NeurIPS'22), and Depts (ICLR'22)). MAE results are shown in the table below. As can be seen, mr-Diff continues to outperform other baselines. It achieves the best results in 10 out of the 12 datasets, and the second-best in the remaining 2 datasets.
>
> |datasets |  mr-Diff  |  TimeDiff |  TimeGrad |  N-Hits |  FiLM |   PatchTST  |   Fedformer |  SCINet |     NLinear|
> | ---- | ---------------------- | ---- | ----- | ----- | ----- | ----- |----- | ----- |----- |
> |tourism\_quarterly |  $\textbf{0.603}$| 0.683| 1.044| 0.703| 0.662| 0.741| 0.716| $\underline{0.651}$| 0.674 |
> |solar\_10\_minutes |  $\textbf{0.230}$| 0.256| 0.328| 0.409| $\underline{0.249}$| 0.329| 0.327| 0.316| 0.298|
> |pedestrian\_counts |  $\textbf{0.114}$| 0.205| 0.220| 0.128| 0.139| 0.137| 0.164| 0.122| $\underline{0.120}$|
> |saugeen\_river\_flow|  $\underline{0.491}$| 0.495| 0.702| 0.622| $\textbf{0.489}$| 0.587| 0.599| 0.536| 0.520|
> |vehicle\_trips |  $\textbf{0.541}$| 0.631| 0.778| 0.566| 0.617| $\underline{0.559}$| 0.628| 0.589| 0.619 |
> |temperature\_rain |  $\textbf{0.474}$| 0.848| 0.970| 0.807| $\underline{0.607}$| 0.890| 1.593| 0.886| 0.921|
> |m4\_yearly |  $\textbf{0.650}$| 0.764| 1.620| $\underline{0.657}$| 0.697| 0.871| 0.679| 0.672| 0.693|
> |m4\_quarterly |  $\textbf{0.306}$| 0.348| 0.657| 0.342| $\underline{0.312}$| 0.677| 0.419| 0.355| 0.341|
> |m4\_monthly |  $\underline{0.294}$| 0.300| 0.490| $\textbf{0.291}$| 0.311| 0.361| 0.376| 0.297| 0.302|
> |m4\_weekly |  $\textbf{0.184}$| 0.200| 0.354| $\underline{0.188}$| 0.251| 0.192| 0.324| 0.189| 0.210|
> |m4\_daily |  $\textbf{0.180}$| 0.195| 0.347| 0.194| $\underline{0.183}$| 0.204| 0.302| 0.198| 0.195|
> |m4\_hourly |  $\textbf{0.211}$| 0.479| 0.846| 0.317| 0.467| 0.299| 0.327| 0.309| $\underline{0.289}$|
>
> **Q5:  Error bars evaluation is not completed (provided for only Electricity, ETTh1, ETTm1 but not the others**
>
> **A5:** As suggested by the reviewer, error bars for the other five datasets are provided in the table below. As can be seen, the largest standard deviation is 0.0023, indicating that the proposed model is robust to different initializations.
>
> ||  NorPool  | | Caiso | | Traffic| | Weather | |  Exchange |  | Wind| |
> | ---- | -----| ---- | ----- | ----- | ----- | ----- |----- | ----- |----- | ----- | ----- | ----- |
> || MAE |  MSE |  MAE |  MSE |  MAE |  MSE |  MAE |  MSE |  MAE |  MSE |  MAE |  MSE |
> |0 |  0.6096 |  0.6668 |  0.2129 |  0.1225 |  0.1973 |  0.1192 |  0.0322 |  0.0019 |  0.0936 |  0.0155 |  1.1704 |  2.1862 |
> |1 |  0.6092 |  0.6664 |  0.2131 |  0.1226 |  0.1974 |  0.1195 |  0.0321 |  0.0018 |  0.0935 |  0.0155 |  1.1682 |  2.1820 |
> |2 |  0.6087 |  0.6659 |  0.2091 |  0.1220 |  0.1970 |  0.1193 |  0.0323 |  0.0018 |  0.0947 |  0.0158 |  1.1690 |  2.1822 |
> |3 |  0.6099 |  0.6672 |  0.2129 |  0.1225 |  0.1976 |  0.1192 |  0.0322 |  0.0019 |  0.0954 |  0.0160 |  1.1676 |  2.1807 |
> |4 |  0.6091 |  0.6665 |  0.2120 |  0.1221 |  0.1973 |  0.1195 |  0.0322 |  0.0018 |  0.0946 |  0.0159 |  1.1670 |  2.1804 |
> |Mean |  0.6093 |  0.6666 |  0.2120 |  0.1223 |  0.1973 |  0.1193 |  0.0322 |  0.0018 |  0.0943 |  0.0157 |  1.1684 |    2.1823 |
> |Std. |  0.0005 |  0.0005 |  0.0017 |  0.0003 |  0.0002 |  0.0001 |  0.0001 |  0.0001 | 0.0008 |  0.0002 |  0.0013 |     0.0023  |

---

> > ### Comment · Reviewer_V8ZH · 2023-11-22
> > **Thank you for addressing my concerns**
> >
> > I appreciate the detailed response from the authors and I would be happy if these discussions and results are included in a revised version. I am keeping my score.
> >
> > For the multiscale Lorenz-96 system, what is the precise equation (and parameters) used for generating the synthetic data and what is the dimension of the system? For the evaluation, it would also be nice to see how various models perform for different values of H (from small to large).

---

> ### Author Response · Authors · 2023-11-21
>
> **Q6: MAE and MSE are used as metrics to evaluate the forecasting performance. How about using MAPE?**
>
> **A6:** It is known that MAPE has limitations [1-2]. One of its main problems is that it is meaningful only for values where divisions and ratios make sense, and produces infinite or undefined values when the actual observations are zero or close to zero. Moreover, it can be biased by the scale of the actual values. For example, suppose you have two observations with the same absolute error, but different actual values. In that case, the one with the smaller actual value will have a larger percentage error and contribute more to the MAPE. Hence, most related time series papers (such as [3-6] below) do not report MAPE.
>
> **References:**
>
> - [1] Tayman J, Swanson D A. On the validity of MAPE as a measure of population forecast accuracy. Population Research and Policy Review, 1999, 18: 299-322.
>
> - [2] Kim S, Kim H. A new metric of absolute percentage error for intermittent demand forecasts. International Journal of Forecasting, 2016, 32(3): 669-679.
>
> - [3] ICML'23 Non-autoregressive conditional diffusion models for time series prediction.
>
> -  [4] ICLR'23 Scaleformer: Iterative multi-scale refining transformers for time series forecasting.
>
> - [5] NeurIPS'22 Generative time series forecasting with diffusion, denoise, and disentanglement.
>
> - [6] ICML'22 FEDformer: frequency enhanced decomposed transformer for long-term series forecasting.
>
> **Q7: How effective is the model for long-term forecasting (i.e. when H is large)? Can it reconstruct the long-term statistics of the underlying dynamical system accurately?**
>
> **A7:** Many of our experiments are on long-term forecasting, e.g., with prediction horizon $H=720$
> (corresponding to forecasting next month) on NorPool and Caiso, and $H=168$ for Traffic and Electricity
> (corresponding to forecasting next week). These $H$ values are usually considered long-term forecasting (see, e.g., [1-3]). As can be seen from Tables 1 and 2, mr-Diff consistently achieves the best average ranking. This demonstrates the effectiveness of mr-Diff for long-term forecasting.
>
> **References:**
>
> - [1] ICML'23 Non-autoregressive conditional diffusion models for time series prediction.
>
> - [2] ICLR'23 Scaleformer: Iterative multi-scale refining transformers for time series forecasting.
>
> - [3] ICML'22 FEDformer: frequency-enhanced decomposed transformer for long-term series forecasting.
>
> **Q8:  What are the limitations of the proposed model?**
>
> **A8:** The main limitation is that the lookback window needs to be sufficiently long and informative. When the lookback window is too short, extracting multiresolution trends becomes challenging, and performance improvement can be less obvious.

---

> ### Author Response · Authors · 2023-11-23
>
> **Q1: I would be happy if these discussions and results were included in a revised version. I am keeping my score.**
>
> **A1:** We would like to sincerely thank Reviewer V8ZH for providing valuable suggestions.
>
> As suggested by the reviewer, we have included the above discussions and results in the revised version.
>
> The main changes in the revised version include:
> - Inclusion of Appendix G, which reports results on the multiscale Lorenz-96 time series for different H's.
> - Addition of Appendix H, which presents the Monash results.
> - Introduction of Tables 12 and 13 in Appendix I, providing a complete error-bar evaluation.
> - Incorporation of Appendix J, which includes discussions regarding the training efficiency and the number of trainable parameters of the proposed model.
>
> **Q2: For the multiscale Lorenz-96 system, what is the precise equation (and parameters) used for generating the synthetic data and what is the dimension of the system?**
>
> **A2:** The precise equation of the multi-scale Lorenz 96 system is
> $$\frac{dX_k}{dt}=X_{k-1}\left(X_{k+1}-X_{k-2}\right)+F-\frac{hc}{b}\Sigma_jY_{j,k},$$
> $$\frac{dY_{j,k}}{dt}=-cbY_{j+1,k}\left(Y_{j+2,k}-Y_{j-1,k}\right)-cY_{j,k}+  \frac{hc}{b}X_k -\frac{he}{d}\Sigma_iZ_{i,j,k},$$
> $$\frac{dZ_{i,j,k}}{dt}=edZ_{i-1,j,k}\left(Z_{i+1,j,k}-Z_{i-2,j,k}\right)- geZ_{i,j,k}+ \frac{he}{d}Y_{j,k}.$$
> Following [1], the values of the parameters are set as follows: $F=20$, $b=c=e=d=g=10$, and $h=1$. The indices $i, j, k$ range from $1$ to $8$, resulting in $X$ having 8 dimensions, while $Y$ and $Z$ have 64 and 512 dimensions, respectively. We utilize the time series data of variable $X$ in the experiments.
>
> **Reference:**
>
> [1] Chattopadhyay A, Subel A, Hassanzadeh P. Data-driven super-parameterization using deep learning: Experimentation with multiscale Lorenz 96 systems and transfer learning[J]. Journal of Advances in Modeling Earth Systems, 2020, 12(11): e2020MS002084.
>
> **Q3: For the evaluation, it would also be nice to see how various models perform for different values of H (from small to large).**
>
> **A3:** The table below summarizes results for different horizons ($H$ in \{24, 96, 168, 336, 720\}). As can be seen, mr-Diff consistently outperforms other baselines across different values of $H$.
> | $H$ |  mr-Diff        | TimeDiff | TimeGrad | N-Hits | FiLM              | PatchTST | Fedformer | SCINet | NLinear           |
> |:------------------:|:--------------:|:--------:|:--------:|:------:|:-----------------:|:--------:|:---------:|:------:|:-----------------:|
> | 24                 | $\textbf{0.260}$   | 0.357    | 0.641    | 0.414  | 0.379             | 0.610    | 0.367     | 0.330  | $\underline{0.265}$ |
> | 96                 | $\textbf{0.730}$ | 0.789    | 0.970    | 0.822  | $\underline{0.761}$ | 0.879    | 0.795     | 0.843  | 0.788             |
> | 168                | $\textbf{0.820}$ | 0.894    | 0.988    | 0.956  | $\underline{0.832}$ | 0.955    | 0.858     | 0.879  | 0.866             |
> | 336                | $\textbf{0.854}$ | 0.938    | 1.011    | 0.987  | $\underline{0.862}$ | 1.055    | 0.897     | 0.989  | 0.958             |
> | 720                | $\textbf{0.903}$ | 1.015    | 1.130    | 1.027  | $\underline{0.933}$ | 1.191    | 0.983     | 1.053  | 1.020             |
> | avg.               | $\textbf{0.714}$ | 0.799    | 0.948    | 0.841  | $\underline{0.753}$ | 0.938    | 0.780     | 0.819  | 0.779             |

---

> ### Comment · Reviewer_V8ZH · 2023-11-23
> **Thank you for the response**
>
> Thank you for the clarifications and the additional results. I am raising my score. Please also include the source code that could reproduce the results in the next revised version.

---

> > ### Author Response · Authors · 2023-11-23
> > **Thank you!**
> >
> > We sincerely appreciate the time you took to review our paper and give us helpful suggestions. Your feedback has greatly improved our work.
> >
> > Thank you for increasing the score!

---

### Official Review · Reviewer_aTA7 · 2023-11-06

**Soundness:** 3 good
**Presentation:** 3 good
**Contribution:** 3 good
**Rating:** 6
**Confidence:** 4

**Summary:**

This paper combines a multi-resolution structure and a diffusion model to forecast a time series. Similar to TimeDiff, it has a forward diffusion process that sequentially extracts fine-to-coarse trends from the time series and a backward denoising process in an easy-to-hard non-autoregressive manner.

**Strengths:**

The main contribution of this work compared to the TimeDiff is that the authors added a multi-resolution structure. The authors have conducted extensive experiments to demonstrate the superiority of the proposed method compared with other existing models.

**Weaknesses:**

The authors mentioned that there have been recent works using U-Net to capture multi-resolution patterns. Considering that the main contribution of this work compared with the TimeDiff is incorporating a multi-resolution structure in a diffusion model, it is essential to compare the proposed seasonal-trend decomposition and the U-Net style decomposition.

In terms of computation complexity, the authors should also compare the model training time, especially with TimeDiff. For inference, the authors stated that "mr-Diff is even faster than TimeDiff" which is also suspicious. All factors presented by the authors apply to TimeDiff as well. Specifically, TimeDiff can be regarded as mr-Diff with S = 1.

There are some claims not well justified.
1. Why “while the seasonal-trend decomposition obtains both the seasonal and trend components, the focus here is on the trend.”?
2. Why “when X0 is used as in existing time series diffusion models, the denoising network may overfit temporal details at the finer level”?

Minor:
unlike existing --> Unlike existing
that directly denoise --> that directly denoises

**Questions:**

See above

---

> ### Author Response · Authors · 2023-11-21
>
> **Q1: The authors mentioned that there have been recent works using U-Net to capture multi-resolution patterns. Considering that the main contribution of this work compared with the TimeDiff is incorporating a multi-resolution structure in a diffusion model, it is essential to compare the proposed seasonal-trend decomposition and the U-Net style decomposition.**
>
> **A1:** There might be some misunderstanding. In our study, we have conducted comparisons with various baselines that use U-Net, including both diffusion methods (TimeGrad, SSSD, and CSDI) and non-diffusion baselines (D$^3$VAE and PSA-GAN). All these baselines utilize U-Net architectures, which consist of multiple residual blocks with downsampling and upsampling to capture multiresolution information in the time series. Additionally, in our ablation studies (refer to $\underline{\textbf{Table 9 in Appendix F}}$), we also compared with a variant of the proposed mr-Diff which replaces the seasonal-trend decomposition with a U-Net style decomposition employing downsampling and upsampling. The results from these comparisons demonstrate the effectiveness of the seasonal-trend decomposition in mr-Diff.
>
> **Q2: In terms of computation complexity, the authors should also compare the model training time, especially with TimeDiff.**
>
> **A2:** As suggested, the table below shows the number of trainable parameters of different time series diffusion models and the corresponding training time (in ms) with different prediction horizons ($H$). As can be seen, the training of mr-Diff is faster than the existing diffusion models.
>
> |      | \# of trainable params | H=96 | H=168 | H=192 | H=336 | H=720 |
> | ---- | :--------------------: | :----: | :-----: | :-----: | :-----: | :-----: |
> | mr-Diff (S=2) | 0.9M | 0.54 | 0.57 | 0.58 | 0.62 | 0.66 |
> | mr-Diff (S=3) | 1.4M | 0.59 | 0.69 | 0.71 | 0.74 | 0.82 |
> | mr-Diff (S=4) | 1.8M |  0.78  |0.96   |0.99   | 1.07   | 1.45 |
> | mr-Diff (S=5) |  2.3M |  1.19     | 1.26   | 1.27   | 1.31   | 1.72 |
> | TimeDiff |  1.7M |  0.71  | 0.75   | 0.77   | 0.82   | 0.85 |
> | TimeGrad |  3.1M |  2.11  | 2.42   | 3.21   | 4.22   | 5.93 |
> | CSDI |  10M |  5.72   | 7.09   | 7.59   | 10.59  | 17.21 |
> | SSSD |  32M |  16.98  | 19.34  | 22.64  | 32.12  | 52.93 |
>
> **Q3: For inference, the authors stated that "mr-Diff is even faster than TimeDiff" which is also suspicious. All factors presented by the authors apply to TimeDiff as well. Specifically, TimeDiff can be regarded as mr-Diff with S = 1.**
>
> **A3:** There might be some misunderstanding. Note that TimeDiff is indeed *NOT* the same as mr-Diff with S = 1, as the conditioning network of mr-Diff is simpler than that of TimeDiff. In TimeDiff, a deep network with 10 convolution layers is used to map the lookback window for future mixup; while mr-Diff simply uses a linear mapping in the conditioning network (see Figure 3). This design choice helps to reduce the computational complexity. Hence, this explains why mr-Diff is faster than TimeDiff also on inference.
>
> **Q4:  Why “while the seasonal-trend decomposition obtains both the seasonal and trend components, the focus here is on the trend.”?**
>
> **A4:** This is because the denoising process of mr-Diff proceeds in an easy-to-hard manner, and thus we focus on the trend to generate fine-to-coarse trends. This is also explained in the last paragraph in Section 4.1: ''*as we use the diffusion model for time series reconstruction at various stages/resolutions (Section 4.2), intuitively, it is easier to predict a finer trend from a coarser trend. Reconstruction of a finer seasonal component from a coarser seasonal component may be difficult, especially as the seasonal component may not present clear patterns*''.
>
> **Q5: Why “when X0 is used as in existing time series diffusion models, the denoising network may overfit temporal details at the finer level”?**
>
> **A5:** As mentioned in the second paragraph of Section 4.2, both the lookback segment $X_s$ and the generation target $Y_s$ are obtained from the same resolution level. Consequently, utilizing $X_s$ facilitates a more effective and straightforward reconstruction of $Y_s$. On the other hand, $X_0$ is obtained at the finest resolution level, and contains temporal details that may not be relevant to $Y_s$. Consequently, including $X_0$ in the denoising process of $Y_s​$ may lead to performance degradation due to overfitting of the unrelated details.
>
>  **Q6: Minor: unlike existing $>$ Unlike existing  that directly denoise $>$ that directly denoises**
>
> **A6:** Thanks for your careful reading. We will correct these typos in the next version.

---

> ### Comment · Reviewer_aTA7 · 2023-12-04
> **Thank you for your response.**
>
> I appreciate the thorough responses and explanations provided by the authors. Most of my comments have been addressed.
> I am not fully convinced that using a finer resolution $X_0$ with better details leads to overfitting unless we have better evidence.
> I will keep the same score and lean towards accepting the paper

---

### Meta-Review · Area_Chair_A77N · 2023-12-12

**Metareview:**

This paper presents a new multi-resolution diffusion model tailored for time series forecasting. Given that generative models for time series remain less explored compared to their counterparts in text and image generation, this work is a relevant contribution. The reviewers recognize the originality of the multi-resolution approach. They also highlight the comprehensive experiments that clearly demonstrate the advantage over other related forecasting models. The authors have effectively addressed questions and concerns during the rebuttal phase, which has further improved the paper. Based on the overall positive feedback from the reviewers, I recommend acceptance of this paper.

**Justification For Why Not Higher Score:**

The paper could be considered for a spotlight presentation. While the methodology is interesting, the practical relevance is limited, i.e., the proposed method is does not substantially outperform all baselines. In light of this I do not recommend spotlight.

**Justification For Why Not Lower Score:**

The proposed method is interesting and relevant for the community. The scores indicate that the paper is above bar.

---

### Decision · Program_Chairs · 2024-01-16

Accept (poster)